# Assessing the Effect of Corporate ESG Management on Corporate Financial & Market Performance and Export

**Oh-Suk Yang [1],\* and Jae-Hoon Han [2],\***

1. Division of Business Administration & Accounting, Kangwon National University, Chuncheon 24341, Republic of Korea
2. Department of Business Administration, Hallym University, Chuncheon 24252, Republic of Korea
\* Correspondence: osyang30@kangwon.ac.kr (O.-S.Y.); jhan@hallym.ac.kr (J.-H.H.)

**Abstract:** The objective of this article is to discover whether a company's ESG management consistently has a positive impact on various corporate performance, such as financial, market and export performance. An empirical analysis employing a fixed effect panel model was conducted using empirical panel data from 2011 to 2021 for 806 non-financial manufacturing and service sector companies in Korea. The main findings are the impact of corporate ESG management on corporate performance varies depending on the type of performance, E and G have a positive effect on corporate profitability, and both positive and negative effects are observed on exports. Regarding market performance, neither ESG was found to have significant effect. The diverse and disproportionate influence of ESG management on financial, market, and export performance presented in this study will provide firms with theoretical and practical implications. However, it is necessary to examine more closely whether these analysis results are the result of actual strategic choices of companies, or a phenomenon in which the level or speed of regulatory and institutional development differs by ESG sector.

**Keywords:** ESG management; financial performance; market performance; export performance; principles for responsible investment; stewardship; stakeholder; financial attributes; Korea





## 1. Introduction

Since James S. Coleman coined the word "social capital" as a concept for measuring corporate value in the 1980s, interest in corporate social responsibility (CSR) and creating shared value (CSV) has gradually increased. More than 30 years have passed since the terms CSR and CSV became hot topics and recently, with the launch of the UN Principles for Responsible Investment (PRI) in 2006, the attention of market actors has shifted to ESG (Environmental, Social, Corporate Governance) management. In particular, as the importance of climate change has emerged and corporate citizenship for the environment, such as carbon neutrality, energy efficiency, and eco-friendly resources, has begun to be emphasized. In addition, the Serious Accidents Punishment Act was recently enforced in South Korea (hereafter Korea), focusing on workplace safety issues, and corporate citizenship is becoming more important to solve social problems that encompass relationships with workers, customers, and the social community. In addition, interest in corporate governance that is focused on resolving agency issues for shareholders' interests, composition of the board of directors, compensation policy, and anti-corruption has begun to lead to corporate participation which allows for mutual growth with partners and resolution of social inequality [1]. This emphasis on ESG management has expanded to the concept of Socially Responsible Investment (SRI) at the global level, and with the growth of ESG funds exceeding $1 trillion in 2020, the size of global investment associated with corporate ESG activities is growing rapidly (about 40 trillion dollars in the same year).

Although Korea has not yet been institutionalized this policy, it is expected that the National Pension Service will invest 50% of the operating fund for domestic Korean

ESG-relevant investment by the year of 2022. In fact, 54% of exporting companies have received CSR evaluation requests from global customers [2], and 20% of domestic small and medium-sized enterprises (SMEs) have experienced ESG evaluation requests in the process of exporting. Most companies answered that it is gradually being strengthened (40.0%) [3]. In addition, in the case of domestic SMEs, 46.8% of the companies responding to the survey were asked to submit certification related to domestic and international ESG certification, fulfilling the request by a self-diagnosis checklist response (31.5%) [3].

What matters is that the sensitivity of ESG factors abroad is higher than in Korea, and it is acting as a cost concept for exporting companies. Exporting companies have faced various non-tariff trade barriers, such as regulations on technology and standards, as well as the European Union's carbon border adjustment system, which is scheduled to be applied to all types of transactions from 2023. Emphasis on ESG management is likely to act as another barrier to exporting companies by putting them at risk of exclusion from selecting partners, reduction in delivery volume, suspension of transactions, and abandonment of exports due to certification costs [4]. As a result, there is a great debate over whether corporate ESG management acts as a cost concept or a benefit concept for companies.

Corresponding to the significance of the issue, various prior studies on ESG management have been conducted, classifying research on ESG metrics [5–7] the impact of ESG activities on financial performance [8–11], determinants of ESG performance [12,13], the relationship between ESG and corporate-level risk or credit rating [14,15] and sustainability research trends [16–18]. Due to the lack of access to detailed data on corporate ESG management activities, these preceding studies have not fully considered the different influences of detailed corporate ESG management activities on various performance types such as financial, market, and export performance. Even previous studies using individual E, S, and G sector data as well as overall ESG ratings [19,20] failed to address the impact on corporate performance by sub-activity information in each sector.

Beyond major disjunctions in the previous research, the guiding research question that this article aims to address is 'Does a company's ESG management consistently have a positive impact on various corporate performances?' In order to find an answer, the authors compared and examined the relationship between ESG management and various types of corporate performance, considering ESG activities in great detail. An empirical analysis using a panel model was conducted using empirical panel data from 2011 to 2021 and, ultimately, the authors' short-cut answer to the research question is not affirmative. The basis for this resolute opinion is as follows. First of all, the impact of corporate ESG management on corporate performance varies depending on the type of performance. In addition, different influences according to the type of corporate performance are observed in each ESG sub-segment. Despite the fact that this study covers a single country case of Korean companies, it is expected that the diverse and disproportionate influence of ESG management on financial, market, and export performance presented in this study will provide theoretical and practical implications at the level of management strategy that companies should fully consider.

This study is structured as follows. Following Section 1, where the research question is suggested, Section 2 explains the theoretical background and related concepts, and presents the main hypotheses to be verified. In Section 3, methodology for research models, data collection, and variable measurement are introduced and in Section 4, the results of empirical analysis on the relationship between ESG management and corporate performance are explained. In Section 5, the theoretical and practical implications derived from this series of reviews are summarized and, finally, in Section 6 conclusions, limitations not covered by this study, and future research themes are presented.

## 2. Theoretical Background and Hypotheses Development

### 2.1. ESG Evaluation Indicators and Major Theories

2.1.1. ESG Evaluation Indicators

In the 1990s, John Elkington's book began to mention the concept of the 'triple bottom line' [21], including financial, environmental, and social factors as the main factors determining corporate value (stock value). The word ESG appeared for the first time in 2003 in the United Nations Environment Program Finance Initiative in the midst of this change in perspective. Later, in 2005, the ESG concept was officially used in the United Nations Global Compact (UNGC), and RE100 on the use of renewable energy declared a plan to replace all energy with renewable energy by 2050. Today, ESG management has come to be used as an investment indicator in the financial market in connection with the concept of social investment, centering on market investors. What matters is that despite the superiority of a certain evaluation agency, the establishment of indicators for the evaluation of corporate ESG management has not yet reached consensus.

There are more than 120 ESG evaluation institutions around the world [22]. CDP, RobecoSAM, Sustainalytics, MSCI, etc., are evaluated as top organizations, and more than 1,000 ESG evaluation indicators are also observed, differing by individual organizations [22]. According to the data of SustainAbility (2020), CDP, RobecoSAM, Sustainalytics, and MSCI are supported with high grades in the evaluation of data quality and data usefulness for various ESG evaluation institutions [23]. As such, different evaluation results can be derived for the same company, so standardization is needed at the global level as well as domestically. If the ESG evaluation standards and evaluation results are different, it is more difficult to broadly apply the significance of ESG management due to problems with consistency and reliability of ESG measurement results.

In spite of the diversity of ESG evaluation indicators, borrowing from ISO 26000, which is commonly used as an index for evaluating corporate sustainability as non-financial performance, a company's ESG management is composed of E, S, and G sectors in detail, which are in turn classified into subcomponents [24]. In Korea, various institutions such as KCGS (Korea Institute of Corporate Governance and Sustainability), Ministry of Commerce, Industry and Energy, and private financial institutions are presenting ESG evaluation indicators. In particular, KCGS has classified ESG evaluation into 7 grades (S, A+, A, B+, B, C, D) and measured every single firms' ESG management using detailed items such as environmental strategy(EST), environmental organization(EOR), environmental management(EMA), environmental performance(EPR), workers(SLA), partners and competitors (SCO), consumers(SCU), local community(SRE), protection of shareholders' rights(GST), board of directors (GBO), disclosure(GLI), audit body(GAU), and distribution of management errors(GDI).

2.1.2. Motivation for ESG management: Stewardship Theory & Stakeholder Perspective

Motivation for companies to achieve social performance through non-financial social activities such as CSR, CSV, and ESG can be divided into the following motives: instrumental, institutional, ethical, and strategic. First of all, from the perspective of instrumental motives, CSR, CSV, and ESG are regarded as tools that give companies competitive advantages [25], enhance investment efficiency [26], provide new business market opportunities, and reduce costs required to respond to new regulations [27]. On the other hand, from the institutional motivation viewpoint, it is emphasized that when a company adapts to a new system, their corporate reputation increases and the overall image of the improved company is projected onto the product image [28]. In terms of ethical motives, companies have a moral obligation to contribute to human development by returning a certain portion of their profits to society [29].

The core theories of strategic management in the area of corporate governance, which constitutes one part of corporate ESG management, are represented by the agency theory and stewardship theory. The agency theory argues that corporate managers act opportunistically to prioritize and maximize their individual interests rather than corporate or

public interests [30,31]. Corporate managers have greater access to corporate information than shareholders, so they are more likely to pursue private interests by using asymmetric information about the company. In addition, corporate managers are likely to make performance-oriented, risk-seeking decisions regardless of whether or not the company will continue to exist and may show interests that conflict with shareholders who have conservative investment tendencies [32].

On the other hand, the stewardship theory holds that from a psychological point of view, corporate managers are motivated by higher-order needs rather than financial needs, and that their private needs can be realized in connection with the organizational purpose to which they belong [33]. Even if there is friction with a heterogeneous group with conflicting interests, corporate managers put the interests of the company first rather than their own interests [34], have an attachment to the organization, identify themselves as part of the company, and pursue the interests of stakeholders at the same time [35,36]. Firms operating under the stewardship theory incur less monitoring costs than those incurring agency costs [37], so the savings from lower costs can be invested in fulfilling social responsibility [38].

The stewardship theory is also embedded in the stakeholder perspective in connection with corporate ESG management. The stakeholder perspective on the motivation for pursuing social performance, which Carroll (1979) argued, is another cross-section of a company's strategic motivation in a sense that a company must satisfy all subjects representing various interests surrounding it [39]. Stakeholders include shareholders [40], business partners [41], customers [42], social movement groups [43], local communities [11], board of directors [26], regulators, policymakers, and standard setters [11]. They have a significant impact on the achievement of corporate social performance and sustainable performance [44]. Therefore, in order for a company to achieve social performance through strategic decision-making and execution such as ESG management, it is necessary to meet the needs of these stakeholders and lead them to reach consensus.

### 2.2. Financial Attributes and Corporate Performance

In the area of environmental uncertainty, firm-specific advantages derived from a given company's financial structure, resources and capabilities, and strategy and system, function as the fundamentals of company heterogeneity. Monopolistic advantages here consist of efficiency, market dominance, easy access to resources, social trust, corporate reputation, and brand image. Monopolistic advantages are firm-specific advantages that provide differentiated capabilities and strategies from competitors to the market, showing the heterogeneity of the company. The relationship between the individual factors that make up firm-specific advantages and the corporate performance is as follows.

#### 2.2.1. Firm Size

The opinions of preceding researchers on the relationship between firm size, a component of monopolistic advantages, and performance are competing without consensus according to performance indicators. Conflicting research results coexist, such as positive (+) relationship between firm size and profitability [45,46], negative (−) relationship with profitability [47], and positive (+) relationship or negative (−) relationship [48,49] with growth performance [50], positive (+) relationship [51,52] or negative (−) with productivity [53]. Despite conflicting views, the authors expect that firm size is a competitive advantage resource based on economies of scale and that, in accordance with firm age, it determines technology and growth strategy for a firm. In addition, firm size, along with specific non-financial management of the firm, is expected to affect firm performance or firm value. Representatively, in a study examining the impact of CSR activities on corporate value by firm size [54], it was confirmed that the impact of specific CSR activities on corporate value was different depending on the firm size. In addition, Shakil (2022) revealed that the effect of ESG management on stock price volatility is controlled by the firm size [55].

**Hypothesis 1.** *Financial characteristics will affect a company's financial performance, market performance, and export performance.*

**Hypothesis 1-1.** *Firm size will have a positive (+) effect on financial performance, market performance, and export performance.*

### 2.2.2. Firm Age

Firm age is likely to have a positive effect on performance indicators through the channels of management competency and marketing competency. First of all, in the case of the management competency channel, companies with a high level of experience have excellent capabilities for effectively managing growth and technology strategies [56]. They may improve profitability by promoting the scale of R&D investment appropriately with excellent coordinating capabilities of resource utilization [57]. In addition, it is possible for them to overcome the time lag (usually 3 to 10 years) between R&D investment and performance linkages [58–61]. It is also easy for them to maintain supplementary competencies that are always available [62]. On the other hand, high firm age through marketing channels is likely to enhance consumers' brand awareness, thereby enhancing consumer loyalty and reliability. This, in turn, can bring a positive signal to market share by providing companies with benefits as market leaders.

**Hypothesis 1-2.** *Firm age will have a positive (+) effect on the company's financial performance, market performance, and export performance.*

### 2.2.3. Financial Soundness (Debt Ratio)

According to previous studies, the debt ratio, which represents a company's financial soundness, has a statistically positive (+) [63] or negative (−) relationship [49,64,65] or no statistically significant relationship [64,66,67]. Summarizing these previous studies, a high debt-to-equity ratio is sometimes understood as a positive sign of a company's ability to raise funds, but it is likely to expand opportunities for operating revenue expense and reduce opportunities for R&D investment. Therefore, it is expected to show a negative relationship with profitability, and low R&D investment activities tend to lower the export competitiveness of the company concerned.

**Hypothesis 1-3.** *Capital adequacy (debt ratio) will affect (−) a company's financial performance, market performance, and export performance.*

### 2.2.4. Current Asset Ratio

The liquidity ratio, which shows short-term debt solvency in relation to a company's financing, is considered as having a positive (+) [66], a negative (−) [45,68] or no statistical significant relationship [44] with a company's profitability. Despite competing views, the authors of this study expect a positive relationship between liquid assets ratio and corporate performance. Companies indirectly or directly raise the necessary funds through banks or the stock market. The capital crunch in the financial market forces banks to engage in deleveraging to collect loan funds in order to maintain capital soundness in accordance with regulations. In this case, the company must have its own financing capabilities, such as a high liquidity ratio.

**Hypothesis 1-4.** *Current asset ratio will have a positive (+) effect on a company's financial performance, market performance, and export performance.*

### 2.2.5. R&D Intensity

At the strategic level, various channels such as external R&D activities; i.e., open innovation in cooperation with external knowledge resources [69,70], or seeking efficiency in knowledge-seeking activities through industry–university cooperation [71] are used for a corporate knowledge management. Although the linkage between internal and external

R&D activities required to sensitively respond to technological changes and challenging market needs is important, the center of a company's knowledge management is its internal R&D activities [72,73]. Through internal R&D activities, companies develop their own capabilities and effectively adapt the organization to environmental dynamics with dynamic capabilities that sense technological changes through continuous monitoring of external R&D activities. On the other hand, excessive R&D investment can reduce corporate profits and threaten short-term survival, so R&D investment on a scale appropriate to the firm size is essential [74]. Therefore, a company's appropriate knowledge management activities start with organizational learning at the organizational competency level [75,76] and progress through knowledge transfer, knowledge sharing, and knowledge creation processes at the system level, resulting in corporate performance.

**Hypothesis 1-5.** *R&D intensity will affect (+) a company's financial performance, market performance, and export performance.*

### 2.2.6. Price-to-Earnings Ratio

The price-to-earnings ratio (PER) is a market value ratio that compares stock price to earnings per share [77]. PER indicates how much greater the stock price is than the earnings per share; that is, it can be said to be an indicator of how the company's stock price is evaluated by the market compared to its net profit [77]. If the PER is low, it means that the stock price is low compared to the company's earnings, and, therefore, the stock price is undervalued compared to the company's market value. The more undervalued a company is, the more likely it is to invest, depending on the type of investor's risk preference [78–81]. In addition, it has been confirmed that companies with high ESG ratings have higher stock return ratios, higher corporate performance and higher Tobin's Q than companies with low ESG ratings [20]. Therefore, the PER can be expected to have a positive relationship with corporate performance.

**Hypothesis 1-6.** *The price-to-earnings ratio will have a positive (+) effect on a company's financial performance, market performance, and export performance.*

### 2.2.7. Price-to-Book Ratio

The price-to-book ratio (PBR) is an indicator that compares the net asset value per share with the stock price [77]. In practice, the market capitalization over net asset calculation formula is often used, and it shows how often the stock price is being traded per share and like PER, it shows the relative level of the stock price. Unlike PER, however, which evaluates stock prices in terms of flow, PBR evaluates companies in terms of stock. Therefore, PER and PBR are complementary to each other and are expected to have a positive relationship with corporate performance.

**Hypothesis 1-7.** *The price-to-book ratio will have a positive (+) effect on the company's financial performance, market performance, and export performance.*

### 2.2.8. Dividend Payout Ratio

A high dividend payout ratio allows shareholders to receive large amount of cash dividends, but if a company distributes a large portion of its profits it can deteriorate the company's financial structure overall [77]. On the other hand, if the dividend payout ratio is maintained at an appropriate level and internal reserves are accumulated, the firm can continue to have growth opportunities through R&D investment.

Proceeding from what has been said above, dividend payout ratio is expected to have a negative relationship with corporate performance.

**Hypothesis 1-8.** *Dividend payout ratio will have a negative (−) effect on the company's financial performance, market performance, and export performance.*

*2.3. Non-Financial Attributes and Corporate Performance*

CSR activities consist of detailed activities such as economic, legal, ethical, and philanthropic responsibilities [38], environmental protection activities, employee satisfaction, win-win cooperation with local communities, and contributions to economic development. These activities increase corporate value and improve reputation, strengthening customers' preference and satisfaction for the company's products and services [82,83]. In contrast, several studies found that CSR activities do not function as corporate investment activities, but end up as costs [84–86]. This discrepancy in the results of previous studies may be a methodological problem, or a problem of CSR measurement methods [87]. While CSR is free from profit maximization by companies doing good deeds, CSV differs in that companies that have joined the market competition try to converge on profit maximization [88]. LVMH's Mecenat project, which restores world cultural heritages, sponsors contemporary art exhibitions, and provides scholarships to young artists, is a representative example of CSR. Marriott Hotel's project to support the poor by providing food and shelter also falls under CSR. Meanwhile, examples of CSV activities include the case of Mitsushita Electric, which developed a non-Freon refrigerator, the case of GlaxoSmithKline, which developed a no-margin meningitis vaccine, and the operation of a UPS youth education program that provided youth with safe driving education.

Since the concepts of ESG, CSR and CSV have some areas in common, the argument about whether CSR and CSV activities are cost concepts or benefit concepts for companies can be applied to ESG activities as well. If ESG is understood as the concept that market investors invest in companies that fulfill their social responsibilities, it is easy to expect that a company's ESG management will improve corporate performance. Contrary to this general opinion, the authors of this study expect that specific non-financial management of companies, such as ESG activities, will have different effects on corporate performance and value. Along the same lines, according to Na and Hong (2011), the relationship between specific CSR activities and corporate values varies depending on the context in which they are interacting [53]. Specifically, a statistically significant negative (-) relationship between environmental protection activities and corporate value (measured by Tobin's Q) was confirmed, and employee satisfaction and contribution to economic development were found to have a positive (+) effect on corporate value.

Unlike CSR, which is not aimed at maximizing corporate profits by escaping from market competition ESG activities are easily projected onto market competition and profit maximization, as they are used as corporate investment indicators. If this logic is applied to the relationship between ESG management and corporate performance, managerial activities in the areas of environment, society, and corporate governance can be expected to show the effect of improving financial performance, market performance, and export performance [19,20]. However, ESG measurement indicators in Korea have not yet reached consensus among the leading institutions and the delay in preparation for ESG management by domestic companies has been observed. Given such an immaturity of institutional and managerial development, it is expected that ESG management is still a cost concept for corporate performance. However, corporate activities in the S domain are expected to improve corporate performance in connection with organizational culture. It is because most sub-components of the S domain cover the internal issues of the company, except for the stakeholder sector.

**Hypothesis 2.** *Non-financial characteristics will affect a company's financial performance, market performance, and export performance.*

**Hypothesis 2-1.** *Environmental Governance will have a negative (−) effect on a company's financial performance, market performance, and export performance.*

**Hypothesis 2-2.** *Social governance will have a positive (+) effect on a company's financial performance, market performance, and export performance.*

**Hypothesis 2-3.** *Corporate governance management activities (Corporate Governance) will have a negative (−) effect on a company's financial performance, market performance, and export performance.*

## 3. Methodology

### 3.1. Data Collection

Using a series of financial or non-financial variables, this study aimed at time-series empirical data from the year 2011 to 2021 for 806 non-financial manufacturing and service sector companies in Korea to examine the determinants of corporate performance. The data from 1001 companies was originally collected, but in the process of collecting additional financial data for every single companies, a total of only 806 companies was selected in the end. Corporate financial data was collected through Bureau van Dijk OSIRIS and TS-2000, and the sectoral data for corporate ESG management was provided by KCGS. Korean public institutions are not officially measuring the ESG management of Korean companies, and the ESG score measured and announced by KCGS is currently being used as a representative ESG measurement value of Korean companies in academic research. Therefore, in the current state, it can be said that the ESG score of KCSG is representative. Stata was used as the statistical package to be used for an analysis.

### 3.2. Research Model

In order to examine the impact of corporate ESG management on exports and financial performance, this study conducted an empirical analysis using financial data and ESG evaluation scores of listed domestic manufacturing companies. The basic regression equation used in the empirical analysis adopted the profitability determinant model of domestic listed companies as a basic model. Specifically, it consists of firm-specific advantages (FSAs) and ESG management attributes. FSAs are subdivided into firm size, firm age, R&D investment, market capitalization, PER, PBR, dividend payout ratio, and financial attributes (liquidity and debt ratio). ESG management attributes are composed of the total score for each E-S-G sector, and the scores for each sub-component in each sector (see Table 1).

**Table 1.** ESG Concepts and Sub-components.

| Sectors | Components | Measure |
|:---:|:---:|:---:|
| E | Environmental Strategy (EST)<br>Environmental Organization (EOR)<br>Environmental Management (EMA)<br>Environmental Performance (EPR)<br>Total Score (ETO) | KCGS Evaluation Data (Score) |
| S | Workers (SLA)<br>Partners and Competitors (SCO)<br>Consumers (SCU)<br>Local Community (SRE)<br>Total Score (STO) | KCGS Evaluation Data (Score) |
| G | Protection of Shareholders' Rights (GST)<br>Board of Directors (GBO)<br>Disclosure (GLI)<br>Audit Body (GAU)<br>Distribution of Management Errors (GDI)<br>Total Score (GTO) | KCGS Evaluation Data (Score) |

Note: E, S and G represent Environmental, Social and Corporate Governance, respectively.

E is composed of environmental strategy, environmental organization, environmental management, environmental performance, and stakeholder response in detail, and S is composed of workers, partners and competitors, consumers, and local communities in detail. G is composed in detail of shareholder rights protection, board of directors, disclosure, audit organization, and distribution of management errors. The final score

for each category was measured by including deduction items common to each category. Meanwhile, dependent variables were composed of export scale, profitability (ROE, ROA), and market performance (Tobin's Q), and the empirical research models, which were divided into basic, reduced, and full models, were adopted as follows:

Model 1 (Basic Model):

$$Performance_{i,t} = \beta_0 + \beta_1 FSA_{i,t} + \varepsilon_{i,t} \tag{1}$$

Model 2 (Reduced Model):

$$Performance_{i,t} = \beta_0 + \beta_1 FSA_{i,t} + \beta_2 E_{i,t} + \beta_3 S_{i,t} + \beta_4 G_{i,t} + \varepsilon_{i,t} \tag{2}$$

Model 3 (Reduced Model):

$$Performance_{i,t} = \beta_0 + \beta_1 FSA_{i,t} + \beta_2 E_{i,j,t} + \beta_3 S_{i,j,t} + \beta_4 G_{i,j,t} + \beta_5 E_{i,t} + \varepsilon_{i,t} \tag{3}$$

Model 4 (Reduced Model):

$$Performance_{i,t} = \beta_0 + \beta_1 FSA_{i,t} + \beta_2 E_{i,j,t} + \beta_3 S_{i,j,t} + \beta_4 G_{i,j,t} + \beta_5 S_{i,t} + \varepsilon_{i,t} \tag{4}$$

Model 5 (Reduced Model):

$$Performance_{i,t} = \beta_0 + \beta_1 FSA_{i,t} + \beta_2 E_{i,j,t} + \beta_3 S_{i,j,t} + \beta_4 G_{i,j,t} + \beta_5 G_{i,t} + \varepsilon_{i,t} \tag{5}$$

Model 6 (Reduced Model):

$$Performance_{i,t} = \beta_0 + \beta_1 FSA_{i,t} + \beta_2 E_{i,j,t} + \beta_3 S_{i,j,t} + \beta_4 G_{i,j,t} + \beta_5 E_{i,t} + \beta_6 S_{i,t} + \beta_7 G_{i,t} + \varepsilon_{i,t} \tag{6}$$

Model 7 (Full Model):

$$Performance_{i,t} = \beta_0 + \beta_1 FSA_{i,t} + \beta_2 E_{i,j,t} + \beta_3 S_{i,j,t} + \beta_4 G_{i,j,t} + \beta_5 E_{i,t} + \beta_6 S_{i,t} + \beta_7 G_{i,t} + \beta_8 TRA_{i,t} + \beta_9 FX_{i,t} + \varepsilon_{i,t} \tag{7}$$

where, *Performance* (Firm Performance), *FSA* (Firm-specific Advantages), $E_{i,t}$ (Total Score of Environmental Sector), $S_{i,t}$ (Total Score of Societal Sector), $G_{i,t}$ (Total Score of Governance Sector), *TRA* (Trade Openness), *FX* (Foreign Exchange Rate), *i* (Individual Firm), *j* (Individual Sub-component in each sector), *t* (Time, Year), $\varepsilon$ (Error Term)

In the regression equation, *i* represents different types of financial institutions, *j* represents an individual sub-component in each sector, and *t* represents time (2011–2021). $Performance_{i,t}$ represents the performance of company *i* at time *t* and at this time, return on equity (ROE), return on total capital (ROA), Tobin's Q and export were adopted as its proxy variables. FSAs stands for firm-specific advantage factors, which include firm size (BS), firm age (AGE), debt ratio (TLTE), current asset ratio (LQA), R&D intensity (RDI), price-to-earnings ratio (PER), and price-to-book ratio (PBR) and dividend payout ratio (POR). The regression equation included non-financial factors composed of the sectoral total scores of ESG and individual sub-components in each sector, as well as the exogenous macroeconomic factors such as trade openness and foreign exchange rate.

A series of reduced models considers the effect of financial attributes and non-financial attributes on corporate performance, respectively. In contrast, in a full model simultaneously considers financial attributes, non-financial attributes and the exogenous macroeconomic factors were simultaneously analyzed. By classifying and examining the basic model, reduced model, and full model, it is possible to compare the fitness of the individual determinant models of corporate performance, and ultimately confirm the relative superiority of the fitness of the full model.

### 3.3. Variable Measurement

3.3.1. Dependent Variables

This study intends to adopt three performance indicators for measuring corporate performance: financial performance, market performance, and export performance (see Table 2). First of all, instead of non-financial performance, which is difficult to objectify and has a wide range of measurement, financial performance with a high level of data availability and comparability was used. Financial performance indicators are divided into profitability, growth potential, stability, and activity. In this study, profitability is intended to be the main indicator of corporate performance. The proxies for profitability that are mainly used include return on assets (ROA) [89–92] and return on equity (ROE) [93–96], net profit margin, operating margin to sales ratio, and total operating profit. Among these indicators, this study adopts ROE as a proxy variable for profitability and ROA as an alternative variable for robustness test. In addition, export size was adopted as a proxy variable for an additional dependent variable to examine the impact on export performance of exporting companies, and Tobin's Q was adopted as a proxy variable to examine the impact on market performance. Tobin's Q can be measured in various ways, but here it was measured as the value by dividing the average market capitalization by the total assets using the simple formula of Li and Wang (2019) [97].

3.3.2. Independent Variables

In this study, independent variables were divided into firm-specific factors as general factors and ESG-specific factors as predictors. Firm-specific factors were further subdivided into firm size, firm age, financial soundness, R&D intensity, and dividend payout ratio. Firm size can be measured by the number of employees [47,98] or by the natural logarithm of total assets. When measured in terms of total assets, it is difficult to rule out methodological errors in which intangible fixed assets are over-measured, depending on the nature of the industry, or tangible fixed assets are underestimated due to depreciation [76]. Therefore, in this study, the number of employees was adopted as a proxy variable for measuring firm size (see Table 2).

Firm age was measured by the number of years of doing business, and financial soundness was measured by the debt ratio (total debt over total equity). To measure short-term debt solvency, which indicates a company's ability to raise funds, the current asset ratio was measured as the ratio of current assets to total capital. R&D intensity is a ratio of R&D expenditure to total sales [99]. The price-to-earnings ratio (PER) PER is a market price of a stock divided by the earnings per share (EPS) that the company will earn [77]. The price-to-book ratio (PBR) is equal to the current stock price divided by the book value per share (BPS), which is the liquidation value of the company [77]. The dividend payout ratio refers to the percentage of a company's net profit paid out as dividends, and is calculated as the total amount of dividends over net profit [77].

3.3.3. Predictor: ESG Management

ESG management evaluation measures, which are non-financial data of companies, use KCGS total score and individual sector scores. The ESG evaluation conducted by KCGS for domestically-listed companies consists of a total of 7 grades (S, A+, A, B+, B, C, D) and scores. Since these 7 ratings are nominal data, some preceding studies use a numerical type converted in reverse order from 7 points to 1 point for empirical analysis. In contrast, this study intends to use the actual measurement values, the sectoral total ESG score and the sub-components' scores for each category (see Table 2).

3.3.4. Control Variables

In this study, macroeconomic indicators related to exports were adopted as control variables to examine export performance. Representatively, dummy variables for individual industries were input to control industry-specific attributes. In addition, exchange rates and trade openness, which are closely related to exports, were added as proxy variables

representing country-specific attributes. For the exchange rate, the annual average monthly exchange rate of USD/KRW was used, and the historical time series data provided by Investing.com was used for the data. Trade openness was collected through the National Statistical Office data service, OECD national data and the IMF database service and was calculated as the ratio of the sum of exports and imports to GDP (see Table 2).

**Table 2.** Variables Definition and Measures.

| Variables Type | Conceptual Definition | Operational Definition (Symbol) | Measures |
|---|---|---|---|
| Dependent Variables | Financial Performance | Profitability | Return on Equity (%) Return on Asset (%) |
| | Market Performance | Tobin's Q | the average market capitalization over total assets |
| | Internationalization | Export | Export amount (natural logarithm) |
| Independent Variables | Firm-specific Advantages | Firm Size (FS) Firm Age (AGE) | Number of employees (natural logarithm) or Total Asset (natural logarithm) Years of doing business |
| | | Financial Soundness: Debt Ratio (TLTE) | |
| | | Current Asset Ratio (LQA) | Total Liabilities over Total Capital (%) Current Asset over Total Capital (%) |
| | | R&D Intensity (RDI) | R&D Expenses over Total Sales (%) |
| | | Price-to-Earnings Ratio (PER) | Annual Highest Price |
| | | Price-to-Book Value Ratio (PBR) | Annual Lowest Price |
| | | Dividend Payout Ratio (POR) | Dividend over Net Income (%) |
| Predictors | ESG Management (E) | Environmental Strategy (EST) Environmental Organization (EOR) Environmental Management (EMA) Environmental Performance (EPR) Total Score (ETO) | |
| | ESG Management (S) | Workers (SLA) Partners and Competitors (SCO) Consumers (SCU) Local Community (SRE) Total Score (STO) | KCGS Evaluation Data (Score) |
| | ESG Management (G) | Protection of Shareholders' Rights (GST) Board of Directors (GBO) Disclosure (GLI) Audit Body (GAU) Distribution of Management Errors (GDI) Total Score (GTO) | |
| Control Variables | Industry-specific Advantages | Industrial Attributes (IND) | Dummy (1~7) |
| | Country-specific Advantages | Trade Openness (TRA) | (Total Export + Total Import) over GDP (%) USD/KRW monthly average |
| | | Foreign Exchange (FX) | |

## 4. Results

### 4.1. Descriptive Statistics

As a result of examining the independent variables representing 8 firm-specific attributes and 16 ESG-related predictors (5 in E, 5 in S, and 6 in G), which were input to the basic empirical analysis model, the number of observations of individual variables was found to be 3271~7143, confirming that there was no excessive disproportionality problem. The observation number of two control variables appeared to be 11,001, indicating a state of equilibrium, and that of the dependent variable was 1347 in the case of export, showing a small number of observations. Therefore, when there is a large difference in the observation values of individual variables, it can be judged that it is unnecessary to conduct winsorization; i.e., to remove the distortion effect of the population by adjusting the necessary scale [100,101].

Table 3 shows the descriptive statistics of the dependent variable, independent variable, and predictors used in the empirical analysis of this study. ROE and ROA, which were used as dependent variables, have standard deviations of 219.6 and 16.07, respectively, between the lowest and highest levels of profitability of Korean companies. In the case of Tobin's Q and export, the standard deviations were 1.55 and 2.48, respectively, indicating that the

difference between companies was not large. Firm size shows a small standard deviation of 1.53, so it can be judged that the firm size is generally even. In the case of firm age, the minimum was 1 year and the maximum was 125 years, but the standard deviation was 19.4, indicating that the difference between companies was not large. The standard deviation of PBR was 6.08, the standard deviation of TRA was 4.37, and the standard deviation of FX was 58.8, showing small figures. The standard deviation of ESG-related variables ranged from a minimum of 0.60 to a maximum of 22.4, showing a generally stable distribution. On the other hand, the standard deviations of the debt ratio and current asset ratio were 305 and 144, respectively, and the standard deviations of R&D concentration, PER, and POR were 2449, 1442, and 1168, respectively, showing significant differences among companies.

**Table 3.** Descriptive Statistics.

| Variables | Obs | Mean | S.D. | Min | Max |
|-----------|-----|------|------|-----|-----|
| ROE | 5985 | 1.20395 | 219.6346 | −8722.31 | 13,840.77 |
| ROA | 5985 | 1.550419 | 16.07561 | −326.926 | 501.3206 |
| TBQ | 5956 | 1.101031 | 1.556617 | 0.001369 | 34.38944 |
| EXP | 1347 | 17.12778 | 2.489176 | 7.600903 | 24.07809 |
| FS | 5986 | 5.853136 | 1.535445 | 0.693147 | 11.67763 |
| AGE | 5986 | 37.90194 | 19.40997 | 1 | 125 |
| TLTE | 5961 | 122.0321 | 305.4166 | 0.05 | 13,769.96 |
| LQA | 5678 | 78.54869 | 144.9453 | −5572.56 | 4303.857 |
| RDI | 3271 | 70.5964 | 2449.343 | $6.57 \times 10^{-06}$ | 129,347.8 |
| PER | 4424 | 104.4057 | 1442.904 | −6.11 | 90,593.87 |
| PBR | 5623 | 3.204492 | 6.082249 | −12.39 | 172.43 |
| POR | 3553 | 71.64879 | 1168.003 | 0.04 | 68,933.92 |
| EST | 7143 | 6.718466 | 6.034193 | 0 | 22 |
| EOR | 7143 | 3.776564 | 3.516003 | 0 | 15 |
| EMA | 7143 | 14.57483 | 14.005 | 0 | 100 |
| EPR | 7143 | 3.842503 | 7.072591 | 0 | 100 |
| ETO | 7143 | 28.78454 | 22.40219 | 0 | 93 |
| SLA | 7131 | 16.63441 | 9.904777 | 0 | 91 |
| SCO | 7131 | 6.470761 | 11.3339 | 0 | 98 |
| SCU | 7131 | 8.432478 | 11.98277 | 0 | 100 |
| SRE | 7131 | 4.282429 | 11.23405 | 0 | 100 |
| STO | 7131 | 27.51522 | 16.79282 | 0 | 93 |
| GST | 6393 | 17.21883 | 10.60273 | 0 | 81 |
| GBO | 6393 | 5.617081 | 5.994008 | 0 | 78 |
| GLI | 6393 | 10.67073 | 14.53046 | 0 | 88 |
| GAU | 6393 | 8.021899 | 8.402006 | 0 | 97 |
| GDI | 6393 | 0.294228 | 0.606798 | 0 | 3 |
| GTO | 6393 | 29.17316 | 10.04905 | 0 | 78 |
| TRA | 11,001 | 35.063 | 4.375138 | 29.795 | 43.07 |
| FX | 11,001 | 1142.557 | 58.88364 | 1054.14 | 1294.16 |

Notes: Obs = Observation; S.D. = Standard Deviation; ROE = Return on Equity; ROA = Return on Assets; TBQ = Tobin's Q; FS = Firm Size; AGE = Firm Age; TLTE = Debt Ratio; LQA = Current Asset Ratio; RDI = R&D Intensity; PER = Price-to-Earnings Ratio; PBR = Price-to-Book Value Ratio; POR = Dividend Payout Ratio; EST = Environmental Strategy; EOR = Environmental Organization; EMA = Environmental Management; EPR = Environmental Performance; ETO = Total Score E; SLA = Workers; SCO = Partners and Competitors; SCU = Consumers; SRE = Local Community; STO = Total Score S; GST = Protection of Shareholders' Rights; GBO = Board of Directors; GLI = Disclosure; GAU = Audit Body; GDI = Distribution of Management Errors; GTO = Total Score G; TRA = Trade Openness; FX = Foreign Exchange.

*4.2. Multicollinearity Test*

If the correlation between independent variables is high, the coefficient estimated by a regression analysis may not have statistical significance. Tables 4–8 show the Pearson correlation and Variance Inflation Factor (VIF) that check the independence of the independent variables used in the statistical empirical analysis conducted in this study. The Pearson's correlation coefficient, which is used to determine the independence of independent variables, does not show a value higher than 0.7, which is the cut-off thresh-

old that confirms multicollinearity of independent variables across all variables [102,103]. Therefore, it can ultimately be judged that there is no multicollinearity problem between independent variables.

In addition, the value of the VIF was checked to supplementarily confirm whether or not multicollinearity, in which one independent variable has endogeneity due to high correlation, is caused by another independent variable. The VIF measures the independence of independent variables by repeatedly performing regression analysis in which a specific independent variable is used as the dependent variable and the remaining variables are selected as independent variables for multiple independent variables. As a result of the check, all independent variables were found to be less than 10 [104], which is the cut-off threshold for determining multicollinearity. Therefore, it can be judged that there is no multicollinearity problem between independent variables. Although there are some variables that exceed the stricter cut-off threshold of 3 [105,106], it is reasonable to use a cut-off threshold of 10 when considering potential linkages between and among ESG sub-components.

**Table 4.** Bivariate Pearson correlation matrix and VIF.

| | VIF | 1 | 2 | 3 | 4 | 5 | 6 | 7 | 8 | 9 | 10 | 11 | 12 | 13 |
|---|---|---|---|---|---|---|---|---|---|---|---|---|---|---|
| 1.ROE | / | 1.00 | | | | | | | | | | | | |
| 2.ROA | / | 0.08 | 1.00 | | | | | | | | | | | |
| 3.TBQ | / | −0.04 | −0.06 | 1.00 | | | | | | | | | | |
| 4.EXP | / | 0.05 | 0.08 | −0.21 | 1.00 | | | | | | | | | |
| 5.FS | 2.32 | 0.02 | 0.15 | −0.21 | 0.51 | 1.00 | | | | | | | | |
| 6.AGE | 1.14 | 0.01 | 0.00 | −0.20 | −0.02 | 0.13 | 1.00 | | | | | | | |
| 7.TLTE | 2.14 | −0.41 | −0.12 | −0.11 | 0.06 | 0.25 | 0.09 | 1.00 | | | | | | |
| 8.LQA | 2.12 | −0.65 | −0.05 | 0.00 | 0.01 | −0.02 | −0.04 | 0.65 | 1.00 | | | | | |
| 9.RDI | 1.17 | 0.00 | −0.05 | 0.08 | −0.18 | −0.03 | −0.03 | −0.01 | 0.00 | 1.00 | | | | |
| 10.PER | 2.59 | −0.02 | −0.02 | 0.07 | −0.07 | −0.02 | 0.01 | −0.02 | −0.03 | 0.03 | 1.00 | | | |
| 11.PBR | 1.19 | −0.39 | −0.19 | 0.60 | −0.13 | −0.16 | −0.13 | 0.23 | 0.35 | 0.02 | 0.08 | 1.00 | | |
| 12.POR | 2.58 | −0.02 | −0.02 | −0.01 | −0.02 | −0.01 | 0.01 | −0.01 | −0.03 | 0.00 | 0.69 | −0.01 | 1.00 | |
| 13.ETO | / | −0.01 | 0.04 | −0.13 | 0.52 | 0.44 | 0.06 | 0.03 | 0.04 | −0.11 | −0.04 | −0.12 | −0.03 | 1.00 |
| 14.STO | / | 0.01 | 0.08 | −0.08 | 0.32 | 0.65 | 0.01 | 0.07 | 0.00 | −0.04 | −0.03 | −0.07 | −0.02 | 0.65 |
| 15.GTO | / | 0.00 | 0.06 | −0.07 | 0.13 | 0.55 | −0.06 | 0.14 | 0.00 | 0.01 | −0.03 | −0.06 | −0.01 | 0.28 |
| 16.EST | 8.64 | 0.00 | 0.02 | −0.08 | 0.30 | 0.18 | 0.01 | 0.01 | 0.04 | −0.08 | −0.02 | −0.07 | −0.02 | 0.61 |
| 17.EOR | 9.67 | 0.00 | 0.04 | −0.07 | 0.29 | 0.20 | 0.04 | 0.01 | 0.04 | −0.09 | −0.02 | −0.06 | −0.02 | 0.64 |
| 18.EMA | 8.10 | −0.01 | 0.01 | −0.12 | 0.43 | 0.35 | 0.06 | 0.01 | 0.04 | −0.08 | −0.03 | −0.10 | −0.02 | 0.69 |
| 19.EPR | 4.64 | 0.00 | 0.00 | −0.09 | 0.25 | 0.34 | 0.03 | 0.04 | 0.01 | −0.05 | −0.02 | −0.07 | −0.01 | 0.42 |
| 20.SLA | 7.48 | −0.01 | 0.04 | −0.11 | 0.36 | 0.46 | 0.05 | 0.03 | 0.01 | −0.08 | −0.03 | −0.09 | −0.02 | 0.61 |

**Table 5.** Bivariate Pearson correlation matrix and VIF.

| | VIF | 14 | 15 | 16 | 17 | 18 | 19 | 20 |
|---|---|---|---|---|---|---|---|---|
| 14.STO | / | 1.00 | | | | | | |
| 15.GTO | / | 0.52 | 1.00 | | | | | |
| 16.EST | 8.64 | 0.46 | 0.18 | 1.00 | | | | |
| 17.EOR | 9.67 | 0.47 | 0.16 | 0.68 | 1.00 | | | |
| 18.EMA | 8.10 | 0.50 | 0.20 | 0.38 | 0.43 | 1.00 | | |
| 19.EPR | 4.64 | 0.38 | 0.25 | −0.02 | 0.01 | 0.62 | 1.00 | |
| 20.SLA | 7.48 | 0.63 | 0.36 | 0.23 | 0.25 | 0.69 | 0.63 | 1.00 |

**Table 6.** Bivariate Pearson correlation matrix and VIF.

| | VIF | 1 | 2 | 3 | 4 | 5 | 6 | 7 | 8 | 9 | 10 | 11 | 12 | 13 | 14 |
|---|---|---|---|---|---|---|---|---|---|---|---|---|---|---|---|
| 21.SCO | 5.29 | 0.00 | 0.02 | −0.07 | 0.21 | 0.46 | 0.01 | 0.04 | 0.01 | 0.00 | −0.01 | −0.06 | −0.01 | 0.33 | 0.59 |
| 22.SCU | 2.89 | 0.02 | 0.00 | −0.05 | 0.08 | 0.32 | 0.04 | 0.07 | −0.02 | 0.00 | −0.01 | −0.04 | −0.01 | 0.22 | 0.42 |
| 23.SRE | 6.17 | 0.00 | 0.01 | −0.06 | 0.23 | 0.43 | 0.00 | 0.04 | −0.01 | −0.02 | −0.02 | −0.06 | −0.01 | 0.22 | 0.42 |
| 24.GST | 6.33 | 0.01 | −0.02 | 0.02 | 0.06 | 0.08 | −0.04 | −0.09 | −0.03 | 0.06 | −0.01 | −0.04 | 0.00 | 0.01 | 0.11 |
| 25.GBO | 5.70 | 0.01 | −0.02 | −0.04 | 0.17 | 0.34 | 0.00 | 0.03 | −0.02 | 0.04 | −0.02 | −0.03 | −0.01 | 0.14 | 0.35 |
| 26.GLI | 6.95 | 0.01 | −0.05 | −0.04 | 0.07 | 0.08 | 0.03 | −0.04 | −0.01 | 0.04 | −0.01 | −0.05 | −0.01 | 0.01 | 0.10 |
| 27.GAU | 5.37 | 0.01 | 0.00 | −0.02 | 0.16 | 0.28 | −0.02 | −0.02 | −0.01 | 0.03 | −0.01 | −0.04 | 0.00 | 0.17 | 0.34 |
| 28.GDI | 6.42 | −0.07 | −0.02 | 0.61 | 0.55 | −0.06 | 0.21 | 0.46 | 0.01 | 0.04 | 0.26 | −0.12 | −0.11 | 0.11 | 0.12 |
| 29.TRA | 2.42 | −0.01 | 0.03 | 0.04 | 0.01 | 0.02 | −0.09 | −0.01 | 0.02 | 0.01 | −0.01 | 0.04 | −0.01 | −0.04 | −0.01 |
| 30.FX | 1.41 | 0.00 | 0.04 | 0.03 | 0.01 | 0.01 | −0.08 | −0.01 | 0.01 | −0.01 | −0.01 | 0.03 | −0.01 | 0.00 | 0.01 |

**Table 7.** Bivariate Pearson correlation matrix and VIF.

| | VIF | 15 | 16 | 17 | 18 | 19 | 20 | 21 | 22 | 23 | 24 | 25 | 26 | 27 |
|---|---|---|---|---|---|---|---|---|---|---|---|---|---|---|
| 21.SCO | 5.29 | 0.34 | −0.02 | −0.01 | 0.58 | 0.65 | 0.65 | 1.00 | | | | | | |
| 22.SCU | 2.89 | 0.22 | −0.09 | −0.07 | 0.49 | 0.60 | 0.50 | 0.61 | 1.00 | | | | | |
| 23.SRE | 6.17 | 0.26 | −0.12 | −0.11 | 0.51 | 0.67 | 0.57 | 0.61 | 0.61 | 1.00 | | | | |
| 24.GST | 6.33 | 0.21 | −0.31 | −0.30 | 0.40 | 0.45 | 0.33 | 0.54 | 0.48 | 0.48 | 1.00 | | | |
| 25.GBO | 5.70 | 0.43 | −0.17 | −0.18 | 0.44 | 0.48 | 0.46 | 0.65 | 0.53 | 0.60 | 0.62 | 1.00 | | |
| 26.GLI | 6.95 | 0.09 | −0.33 | −0.32 | 0.43 | 0.48 | 0.35 | 0.57 | 0.51 | 0.50 | 0.64 | 0.62 | 1.00 | |
| 27.GAU | 5.37 | 0.36 | −0.11 | −0.11 | 0.45 | 0.47 | 0.46 | 0.69 | 0.54 | 0.67 | 0.60 | 0.64 | 0.55 | 1.00 |
| 28.GDI | 6.42 | 0.37 | 0.16 | 0.14 | 0.00 | −0.02 | 0.05 | −0.06 | −0.07 | −0.09 | −0.03 | −0.04 | −0.15 | 0.00 |
| 29.TRA | 2.42 | 0.35 | 0.12 | 0.02 | −0.18 | −0.16 | −0.08 | −0.19 | −0.24 | −0.20 | −0.19 | −0.15 | −0.34 | −0.07 |
| 30.FX | 1.41 | 0.20 | 0.14 | 0.07 | −0.15 | −0.17 | −0.08 | −0.19 | −0.21 | −0.19 | −0.23 | −0.17 | −0.36 | −0.05 |

**Table 8.** Bivariate Pearson correlation matrix and VIF.

| | VIF | 28 | 29 | 30 |
|---|---|---|---|---|
| 28.GDI | / | 1.00 | | |
| 29.TRA | / | 0.52 | 1.00 | |
| 30.FX | / | 0.31 | 0.66 | 1.00 |

Notes: ROE = Return on Equity; ROA = Return on Assets; TBQ = Tobin's Q; FS = Firm Age; TLTE = Debt Ratio; LQA = Current Asset Ratio; RDI = R&D Intensity; PER = Price-to-Earnings Ratio; PBR = Price-to-Book Value Ratio; POR = Dividend Payout Ratio; EST = Environmental Strategy; EOR = Environmental Organization; EMA = Environmental Management; EPR = Environmental Performance; ETO = Total Score E; SLA = Workers; SCO = Partners and Competitors; SCU = Consumers; SRE = Local Community; STO = Total Score S; GST = Protection of Shareholders' Rights; GBO = Board of Directors; GLI = Disclosure; GAU = Audit Body; GDI = Distribution of Management Errors; GTO = Total Score G; TRA = Trade Openness; FX = Foreign Exchange.

*4.3. Adequacy of Research Model*

4.3.1. Unit Root Test

Empirical analysis using panel time series data involves the F-test to check the comparative fit between the OLS model and the panel model, and the Hausman test to compare the fit between the fixed effect model and the random effect model. In addition, before these two tests, a unit root test to check the stability of the time series data used in this study is needed [107]. There are various unit root test methods such as ADF (Augmented Dickey-Fuller), LLC (Levin-Lin-Chu), HT (Harris-Tzavalis), and IPS (Im-Pesaran-Shin). Among these, this study adopted the Fisher-type test to solve the auto-correlation problem for the error term. The Fisher-type test follows the ADF method, which represents the highest stringency in panel model analysis. As a result of the unit root test on the time series empirical data, both the dependent variable and the predictors rejected the null hypothesis that "there is a panel unit root" at the statistical significance level of 1%; i.e., *p*-value < 0.01, confirming the stability of the time series data without spurious regression

problems (See Table 9.). Therefore, the final research model can be selected through the F-test and the Hausman test using time series data without applying the difference model.

**Table 9.** Result of Unit Root Test.

| Variables | Statistic | *p*-Value | Variables | Statistic | *p*-Value |
|---|---|---|---|---|---|
| ROE | $1.33 \times 10^{04}$ | 0.0000 | EPR | 2140.461 | 0.0000 |
| ROA | $1.33 \times 10^{04}$ | 0.0000 | ETO | 2386.101 | 0.0000 |
| TBQ | $1.33 \times 10^{04}$ | 0.0043 | SLA | 2822.816 | 0.0000 |
| EXP | $1.33 \times 10^{04}$ | 0.0070 | SCO | 2304.631 | 0.0000 |
| FS | 123.0641 | 0.0032 | SCU | 1707.153 | 0.0000 |
| AGE | 375.8066 | 0.0000 | SRE | 1172.909 | 0.0000 |
| TLTE | 9118.231 | 0.0000 | STO | 1787.132 | 0.0000 |
| LQA | $1.03 \times 10^{04}$ | 0.0000 | GST | 2009.453 | 0.0000 |
| RDI | 6279.874 | 0.0000 | GBO | 1079.521 | 0.0000 |
| PER | 9903.014 | 0.0000 | GLI | 4946.982 | 0.0000 |
| PBR | 1734.216 | 0.0000 | GAU | 1261.200 | 0.0000 |
| POR | 5846.169 | 0.0000 | GDI | 992.3419 | 0.0000 |
| EST | 2417.237 | 0.0000 | GTO | 1246.595 | 0.0000 |
| EOR | 782.0469 | 0.0000 | TRA | 572.9298 | 0.0000 |
| EMA | 1004.337 | 0.0000 | FX | 387.3287 | 0.0000 |

Notes: ROE = Return on Equity; ROA = Return on Assets; TBQ = Tobin's Q; FS = Firm Size; AGE = Firm Age; TLTE = Debt Ratio; LQA = Current Asset Ratio; RDI = R&D Intensity; PER = Price-to-Earnings Ratio; PBR = Price-to-Book Value Ratio; POR = Dividend Payout Ratio; EST = Environmental Strategy; EOR = Environmental Organization; EMA = Environmental Management; EPR = Environmental Performance; ETO = Total Score E; SLA = Workers; SCO = Partners and Competitors; SCU = Consumers; SRE = Local Community; STO = Total Score S; GST = Protection of Shareholders' Rights; GBO = Board of Directors; GLI = Disclosure; GAU = Audit Body; GDI = Distribution of Management Errors; GTO = Total Score G; TRA = Trade Openness; FX = Foreign Exchange.

### 4.3.2. F-test and Hausman Test

For hypotheses testing, this study first checked the possibility of a first-order auto-correlation (AR (1)) problem for the error term of the basic regression equation in order to find an efficient estimation model through a validity test between Pooled OLS and the panel model. The panel model is further divided into a fixed effect model and a random effect model. For the validity test, two tests are used to find an efficient estimation model. One is the F-test to check the validity between the joint least squares method and the fixed-effects model, and the other is the Breusch-Pagan test to check the validity between the joint least squares method and the random effects model. As a result of the F-test, the relatively optimal fit of the fixed-effects model was confirmed, and the Breusch-Pagan-test confirmed the relatively optimal fit of the random-effects model. In addition, the Hausman test was conducted to confirm the validity between the fixed-effects model and the random-effects model. As a result of the Hausman test, the null hypothesis ($H_0$: cov ($X_{it}$, $u_i$) = 0) that individual independent variables do not correlate with the error term was rejected at the significance level of 1%, confirming that the fixed-effects model is the most efficient estimation method.

### 4.4. Estimation Result

Tables 10–13 show the results of the empirical analysis by applying the fixed-effects panel model to 806 domestic companies by basic, reduced, and full models. The analysis results center on the full model with a relatively excellent $R^2$ value. To summarize the results after inputting all independent variables and predictors: first of all, it was confirmed that financial attributes such as firm size (−), debt ratio (−), PER (+), PBR (+), and POR (−) act as statistically significant determinants of ROE. Second, among ESG activities, EPR (+) and GBO (+) showed a statistically significant relationship with corporate performance, while GDI was dropped. All of the other ESG sub-components were confirmed to have no statistically significant relationship with corporate performance.

Regarding the dependent variable ROA, the same results as ROE were obtained in the case of general financial attributes, and in the case of ESG activities, EOR (+) and GBO (+) showed statistically significant relationships with corporate performance, and GDI was dropped. In the case of the dependent variable Tobin's Q, financial attributes such as TLTE (−), LQA (+), RDI (+), and PBR (+) showed statistically significant relationships, and ESG activities all showed statistically non-significant relationships with corporate performance. In the case of the dependent variable export, all general financial attributes showed no statistically significant relationship with corporate performance, and EST (+), EOR (−), and SRE (−) showed statistically significant relationships while GDI was dropped.

Based on the fact that the adequacy of the fixed-effects model is proven, in the case of a research model with ROE and ROA as dependent variables, hypotheses 1-3 (TLTE), 1-6 (PER), 1-7 (PBR), and 1-8 (POR) were supported, and hypotheses 1-1 (FS), 1-2 (AGE), 1-4 (LQA), and 1-5 (RDI) were not supported. Meanwhile, in the research model with Tobin's Q as a dependent variable, only hypotheses 1-3 (TLTE), 1-4 (LQA), 1-5 (RDI), and 1-7 (PBR) were supported, and the other hypotheses were not supported. In the research model with export as a dependent variable, all hypotheses related to firm-specific factors were not supported. Thereby, corporate financial information partially showed a statistically significant relationship with accounting financial performance indicators and market value indicators. This result presents the evidence that financial information disclosure improves or deteriorates a company's financial and market performance. On the other hand, financial information does not show a statistically significant relationship with export performance, so it can be interpreted that the export market does not respond to a company's financial information and that financial information is not meaningful for corporate export.

Among the hypotheses on non-financial ESG management, Hypothesis 2-2 (social business activities) and Hypothesis 2-3 (Corporate Governance Structure) were not supported across the financial performance research model, market performance research model, and export performance research model. Meanwhile, Hypothesis 2-1 (Environmental Management Activities) was partially supported. Although some hypotheses showed statistical significance, contrary to the authors' expectations, the direction of the relationship was different, leading hypotheses to be unsupported. In summary, with respect to accounting financial performance indicators and export performance indicators, non-financial information such as ESG showed a partially statistically significant relationship, thereby finding evidence that non-financial information disclosure improves or worsens a company's financial and export performance. This fact supports both the claim that corporate spending for ESG management acts as a cost to the company [108,109] and that it enhances corporate competitiveness [110,111]. On the other hand, non-financial information did not show a statistically significant relationship with corporate value such as Tobin's Q, showing the contradictory results with some previous studies [19,20].

Therefore, it can be interpreted that the capital market does not respond to corporate non-financial information and that non-financial information is not meaningful to investors. As a result, the assertion that corporate expenditure for ESG has an asset character that enhances corporate value [19,20,110,112] was not supported. Neither did the negative viewpoint about the relationship between ESG management and corporate value [84,109].

### 4.5. Robustness Test

The results of the empirical analysis conducted in this study need to ensure consistency through robustness testing. There are various tools used for robustness verification but in this study, three methods were adopted: the sequential control variable input method, substitution with other proxy variables, and the Hansen Test to check coefficient stability. First of all, in order to confirm the spatial dependence between data that causes bias in the estimation results [113], control variables were input sequentially rather than simultaneously to confirm the results. As a result of statistical analysis according to sequential input, there was no significant change in the significance level of individual variables, and the causality direction of regression estimation coefficients did not change. For the second

robustness test, the number of employees (natural logarithm) was substituted for total assets (natural logarithm) as a proxy variable for firm size, and short-term capital over total liabilities (%) was substituted for asset quality to ensure consistency of empirical analysis results. As a result of the verification, in the results of the empirical analysis before and after substitution, the statistical significance level and coefficient did not show a significant difference in the causality between the variable and the dependent variable. Lastly, during the entire sample period (2011–2021) for data collection used in the empirical analysis of this article, the Hansen test was conducted to check the stability of the constant term and estimated coefficients of the time series data according to the panel model analysis. As a result of the test, the null hypothesis, i.e. "the coefficient is stable", was not rejected based on the critical value presented by Hansen (1992), so it can be judged that the estimated coefficient derived from the empirical analysis of this study is stable [114].

**Table 10.** Fixed effect panel model estimation of the impact of ESG management on ROE.

|  | Model 1 | Model 2 | Model 3 | Model 4 | Model 5 | Model 6 | Model 7 |
|---|---|---|---|---|---|---|---|
| FS | −1.472 | −11.458 *** | −11.766 *** | −11.689 *** | −12.356 *** | −12.501 *** | −12.498 *** |
|  | (1.583) | (3.464) | (3.454) | (3.482) | (3.480) | (3.500) | (3.507) |
| AGE | 0.040 | 0.342 | 0.553 | 0.336 | 0.465 | 0.568 | 0.384 |
|  | (0.149) | (0.379) | (0.412) | (0.406) | (0.505) | (0.532) | (0.842) |
| TLTE | −0.082 *** | −0.067 ** | −0.066 ** | −0.067 ** | −0.067 ** | −0.069 ** | −0.070 ** |
|  | (0.015) | (0.030) | (0.030) | (0.031) | (0.030) | (0.031) | (0.031) |
| LQA | 0.110 *** | 0.025 | 0.024 | 0.027 | 0.030 | 0.029 | 0.031 |
|  | (0.021) | (0.041) | (0.041) | (0.041) | (0.041) | (0.041) | (0.041) |
| RDI | −0.414 *** | −0.517 | −0.404 | −0.531 | −0.497 | −0.408 | −0.411 |
|  | (0.149) | (0.334) | (0.335) | (0.335) | (0.332) | (0.340) | (0.340) |
| PER | 0.000 | 0.008 *** | 0.008 *** | 0.008 *** | 0.008 *** | 0.008 *** | 0.008 *** |
|  | (0.001) | (0.002) | (0.002) | (0.002) | (0.002) | (0.002) | (0.002) |
| PBR | 0.560 *** | 0.775 ** | 0.695 ** | 0.729 ** | 0.697** | 0.681 * | 0.662 * |
|  | (0.131) | (0.349) | (0.349) | (0.350) | (0.349) | (0.353) | (0.355) |
| POR | −0.004 ** | −0.039 *** | −0.039 *** | −0.039 *** | −0.040 *** | −0.039 *** | −0.039 *** |
|  | (0.002) | (0.008) | (0.008) | (0.008) | (0.008) | (0.008) | (0.008) |
| EST |  |  | −0.269 |  |  | 0.000 | −0.004 |
|  |  |  | (0.224) |  |  | (0.096) | (0.097) |
| EOR |  |  | 0.847 ** |  |  | 0.005 | 0.011 |
|  |  |  | (0.347) |  |  | (0.054) | (0.056) |
| EMA |  |  | −0.006 |  |  | −0.301 | −0.301 |
|  |  |  | (0.072) |  |  | (0.234) | (0.234) |
| EPR |  |  | 0.084 |  |  | 0.865 ** | 0.877 ** |
|  |  |  | (0.072) |  |  | (0.361) | (0.367) |
| ETO |  | 0.080 | 0.005 |  |  | −0.027 | −0.020 |
|  |  | (0.056) | (0.088) |  |  | (0.095) | (0.096) |
| SLA |  |  |  | −0.026 |  | 0.072 | 0.081 |
|  |  |  |  | (0.048) |  | (0.086) | (0.088) |
| SCO |  |  |  | −0.008 |  | 0.030 | 0.023 |
|  |  |  |  | (0.047) |  | (0.093) | (0.096) |
| SCU |  |  |  | 0.007 |  | 0.015 | 0.014 |
|  |  |  |  | (0.031) |  | (0.054) | (0.054) |
| SRE |  |  |  | 0.005 |  | 0.019 | 0.019 |
|  |  |  |  | (0.042) |  | (0.034) | (0.034) |
| STO |  | −0.003 |  | 0.019 |  | −0.010 | −0.015 |
|  |  | (0.041) |  | (0.045) |  | (0.049) | (0.050) |
| GST |  |  |  |  | 0.011 | 0.009 | 0.024 |
|  |  |  |  |  | (0.064) | (0.068) | (0.074) |

**Table 10.** *Cont.*

|  | Model 1 | Model 2 | Model 3 | Model 4 | Model 5 | Model 6 | Model 7 |
|---|---|---|---|---|---|---|---|
| GBO |  |  |  |  | 0.214 * | 0.226 * | 0.232 * |
|  |  |  |  |  | (0.123) | (0.124) | (0.124) |
| GLI |  |  |  |  | −0.084 * | −0.076 | −0.072 |
|  |  |  |  |  | (0.047) | (0.051) | (0.052) |
| GAU |  |  |  |  | −0.040 | −0.059 | −0.060 |
|  |  |  |  |  | (0.052) | (0.067) | (0.068) |
| GTO |  | 0.028 |  |  | 0.034 | 0.033 | 0.029 |
|  |  | (0.078) |  |  | (0.080) | (0.082) | (0.082) |
| TRA |  |  |  |  |  |  | 0.253 |
|  |  |  |  |  |  |  | (0.451) |
| FX |  |  |  |  |  |  | −0.005 |
|  |  |  |  |  |  |  | (0.015) |
| Industry dummies | YES | YES | YES | YES | YES | YES | YES |
| _cons | 45.078 | 307.790 *** | 309.176 *** | 317.541 *** | 329.760 *** | 327.370 *** | 331.304 *** |
|  | (39.622) | (88.390) | (88.118) | (88.842) | (89.071) | (89.850) | (96.442) |
| N | 1798.000 | 1029.000 | 1031.000 | 1031.000 | 1029.000 | 1029.000 | 1029.000 |
| $R^2$ | 0.055 | 0.101 | 0.111 | 0.098 | 0.106 | 0.118 | 0.119 |
| Hausman (Prob > chi2) | 0.0000 | 0.0000 | 0.0000 | 0.0000 | 0.0000 | 0.0000 | 0.0000 |

Notes: ***, **, and * refer to significance at 1%, 5%, and 10%, respectively; GDI has been dropped; ROE = Return on Equity; ROA = Return on Assets; TBQ = Tobin's Q; FS = Firm Size; AGE = Firm Age; TLTE = Debt Ratio; LQA = Current Asset Ratio; RDI = R&D Intensity; PER = Price-to-Earnings Ratio; PBR = Price-to-Book Value Ratio; POR = Dividend Payout Ratio; EST = Environmental Strategy; EOR = Environmental Organization; EMA = Environmental Management; EPR = Environmental Performance; ETO = Total Score E; SLA = Workers; SCO = Partners and Competitors; SCU = Consumers; SRE = Local Community; STO = Total Score S; GST = Protection of Shareholders' Rights; GBO = Board of Directors; GLI = Disclosure; GAU = Audit Body; GDI = Distribution of Management Errors; GTO = Total Score G; TRA = Trade Openness; FX = Foreign Exchange.

**Table 11.** Fixed effect panel model estimation of the impact of ESG management on ROA.

|  | Model 8 | Model 9 | Model 10 | Model 11 | Model 12 | Model 13 | Model 14 |
|---|---|---|---|---|---|---|---|
| FS | −2.322 * | −12.263 *** | −12.544 *** | −12.381 *** | −13.030 *** | −13.055 *** | −13.053 *** |
|  | (1.270) | (3.083) | (3.073) | (3.097) | (3.096) | (3.113) | (3.120) |
| AGE | 0.062 | 0.428 | 0.623 * | 0.475 | 0.477 | 0.577 | 0.498 |
|  | (0.120) | (0.337) | (0.367) | (0.361) | (0.449) | (0.473) | (0.749) |
| TLTE | −0.074 *** | −0.064 ** | −0.063 ** | −0.065** | −0.064 ** | −0.066 ** | −0.066 ** |
|  | (0.012) | (0.027) | (0.027) | (0.027) | (0.027) | (0.027) | (0.027) |
| LQA | 0.063 *** | 0.024 | 0.022 | 0.025 | 0.027 | 0.026 | 0.026 |
|  | (0.017) | (0.036) | (0.036) | (0.036) | (0.037) | (0.037) | (0.037) |
| RDI | −0.265 ** | −0.312 | −0.203 | −0.323 | −0.286 | −0.198 | −0.199 |
|  | (0.120) | (0.297) | (0.298) | (0.298) | (0.295) | (0.302) | (0.303) |
| PER | 0.000 | 0.006 *** | 0.006 *** | 0.006 *** | 0.006 *** | 0.006 *** | 0.006 *** |
|  | (0.001) | (0.002) | (0.002) | (0.002) | (0.002) | (0.002) | (0.002) |
| PBR | 0.364 *** | 0.402 | 0.337 | 0.367 | 0.328 | 0.315 | 0.307 |
|  | (0.105) | (0.310) | (0.310) | (0.311) | (0.311) | (0.314) | (0.316) |
| POR | −0.003 ** | −0.029 *** | −0.029 *** | −0.029 *** | −0.029 *** | −0.029 *** | −0.029 *** |
|  | (0.001) | (0.007) | (0.007) | (0.007) | (0.007) | (0.007) | (0.007) |
| EST |  |  | −0.230 |  |  | −0.228 | −0.228 |
|  |  |  | (0.199) |  |  | (0.208) | (0.209) |

**Table 11.** *Cont.*

| | Model 8 | Model 9 | Model 10 | Model 11 | Model 12 | Model 13 | Model 14 |
|---|---|---|---|---|---|---|---|
| EOR | | | 0.811 *** | | | 0.868 *** | 0.872 *** |
| | | | (0.309) | | | (0.321) | (0.327) |
| EMA | | | 0.023 | | | 0.015 | 0.018 |
| | | | (0.064) | | | (0.085) | (0.086) |
| EPR | | | 0.056 | | | 0.052 | 0.055 |
| | | | (0.064) | | | (0.077) | (0.078) |
| ETO | | 0.057 | −0.021 | | | −0.046 | −0.048 |
| | | (0.049) | (0.078) | | | (0.085) | (0.086) |
| SLA | | | | −0.006 | | 0.042 | 0.039 |
| | | | | (0.043) | | (0.083) | (0.085) |
| SCO | | | | −0.016 | | −0.002 | −0.002 |
| | | | | (0.042) | | (0.048) | (0.048) |
| SCU | | | | 0.004 | | 0.015 | 0.015 |
| | | | | (0.028) | | (0.030) | (0.030) |
| SRE | | | | 0.005 | | −0.001 | −0.003 |
| | | | | (0.038) | | (0.043) | (0.044) |
| STO | | 0.002 | | 0.019 | | 0.009 | 0.011 |
| | | (0.036) | | (0.040) | | (0.048) | (0.050) |
| GST | | | | | 0.008 | 0.008 | 0.014 |
| | | | | | (0.057) | (0.061) | (0.066) |
| GBO | | | | | 0.214 * | 0.231 ** | 0.233 ** |
| | | | | | (0.109) | (0.110) | (0.111) |
| GLI | | | | | −0.069 * | −0.065 | −0.063 |
| | | | | | (0.042) | (0.046) | (0.046) |
| GAU | | | | | −0.057 | −0.080 | −0.080 |
| | | | | | (0.046) | (0.060) | (0.060) |
| GTO | | 0.033 | | | 0.041 | 0.045 | 0.044 |
| | | (0.069) | | | (0.071) | (0.073) | (0.073) |
| TRA | | | | | | | 0.101 |
| | | | | | | | (0.401) |
| FX | | | | | | | −0.002 |
| | | | | | | | (0.014) |
| Industry dummies | Yes | Yes | Yes | Yes | Yes | Yes | Yes |
| _cons | 67.122 ** | 324.010 *** | 325.048 *** | 327.889 *** | 344.544 *** | 339.032 *** | 340.922 *** |
| | (31.796) | (78.665) | (78.409) | (79.025) | (79.228) | (79.906) | (85.786) |
| N | 1798.000 | 1029.000 | 1031.000 | 1031.000 | 1029.000 | 1029.000 | 1029.000 |
| R$^2$ | 0.053 | 0.092 | 0.102 | 0.090 | 0.098 | 0.110 | 0.111 |
| Hausman (Prob > chi2) | 0.0000 | 0.0000 | 0.0000 | 0.0000 | 0.0000 | 0.0000 | 0.0000 |

Notes: ***, **, and * refer to significance at 1%, 5%, and 10%, respectively; GDI has been dropped; ROE = Return on Equity; ROA = Return on Assets; TBQ = Tobin's Q; FS = Firm Size; AGE = Firm Age; TLTE = Debt Ratio; LQA = Current Asset Ratio; RDI = R&D Intensity; PER = Price-to-Earnings Ratio; PBR = Price-to-Book Value Ratio; POR = Dividend Payout Ratio; EST = Environmental Strategy; EOR = Environmental Organization; EMA = Environmental Management; EPR = Environmental Performance; ETO = Total Score E; SLA = Workers; SCO = Partners and Competitors; SCU = Consumers; SRE = Local Community; STO = Total Score S; GST = Protection of Shareholders' Rights; GBO = Board of Directors; GLI = Disclosure; GAU = Audit Body; GDI = Distribution of Management Errors; GTO = Total Score G; TRA = Trade Openness; FX = Foreign Exchange.

**Table 12.** Fixed effect panel model estimation of the impact of ESG management on Tobin's Q.

| | Model 15 | Model 16 | Model 17 | Model 18 | Model 19 | Model 20 | Model 21 |
|---|---|---|---|---|---|---|---|
| FS | −0.245 *** | −0.089 | −0.087 | −0.077 | −0.090 | −0.073 | −0.088 |
| | (0.091) | (0.126) | (0.127) | (0.126) | (0.128) | (0.127) | (0.126) |
| AGE | 0.003 | −0.052 *** | −0.057 *** | −0.057 *** | −0.056 *** | −0.077 *** | −0.045 |
| | (0.009) | (0.014) | (0.015) | (0.015) | (0.019) | (0.019) | (0.030) |
| TLTE | −0.006 *** | −0.006 *** | −0.006 *** | −0.006 *** | −0.006 *** | −0.006 *** | −0.006 *** |
| | (0.001) | (0.001) | (0.001) | (0.001) | (0.001) | (0.001) | (0.001) |
| LQA | 0.005 *** | 0.004 *** | 0.005 *** | 0.005 *** | 0.005 *** | 0.005 *** | 0.005 *** |
| | (0.001) | (0.001) | (0.001) | (0.001) | (0.002) | (0.002) | (0.001) |
| RDI | −0.013 | 0.022 * | 0.017 | 0.021 * | 0.017 | 0.021 * | 0.021 * |
| | (0.009) | (0.012) | (0.012) | (0.012) | (0.012) | (0.012) | (0.012) |
| PER | −0.000 | 0.000 | 0.000 | 0.000 | 0.000 | 0.000 | 0.000 |
| | (0.000) | (0.000) | (0.000) | (0.000) | (0.000) | (0.000) | (0.000) |
| PBR | 0.152 *** | 0.194 *** | 0.196 *** | 0.197 *** | 0.199 *** | 0.196 *** | 0.194 *** |
| | (0.007) | (0.013) | (0.013) | (0.013) | (0.013) | (0.013) | (0.013) |
| POR | 0.000 | −0.000 | −0.000 | −0.000 | −0.000 | | |
| | (0.000) | (0.000) | (0.000) | (0.000) | (0.000) | | |
| EST | | | −0.004 | | | −0.005 | −0.003 |
| | | | (0.008) | | | (0.009) | (0.008) |
| EOR | | | −0.008 | | | −0.011 | −0.001 |
| | | | (0.013) | | | (0.013) | (0.013) |
| EMA | | | −0.002 | | | 0.001 | 0.003 |
| | | | (0.003) | | | (0.003) | (0.003) |
| EPR | | | 0.002 | | | 0.003 | 0.004 |
| | | | (0.003) | | | (0.003) | (0.003) |
| ETO | | −0.002 | −0.001 | | | 0.000 | −0.001 |
| | | (0.002) | (0.003) | | | (0.003) | (0.003) |
| SLA | | | | 0.001 | | −0.004 | −0.003 |
| | | | | (0.002) | | (0.003) | (0.003) |
| SCO | | | | −0.000 | | −0.002 | −0.002 |
| | | | | (0.002) | | (0.002) | (0.002) |
| SCU | | | | 0.001 | | 0.000 | 0.000 |
| | | | | (0.001) | | (0.001) | (0.001) |
| SRE | | | | 0.000 | | −0.000 | −0.001 |
| | | | | (0.002) | | (0.002) | (0.002) |
| STO | | −0.004 *** | | −0.005 *** | | −0.003 * | −0.002 |
| | | (0.001) | | (0.002) | | (0.002) | (0.002) |
| GST | | | | | 0.003 | 0.002 | 0.004 |
| | | | | | (0.002) | (0.002) | (0.003) |
| GBO | | | | | 0.001 | 0.000 | 0.002 |
| | | | | | (0.004) | (0.005) | (0.004) |
| GLI | | | | | −0.001 | −0.001 | −0.000 |
| | | | | | (0.002) | (0.002) | (0.002) |
| GAU | | | | | −0.001 | 0.000 | −0.001 |
| | | | | | (0.002) | (0.002) | (0.002) |
| GTO | | 0.003 | | | 0.001 | 0.002 | 0.000 |
| | | (0.003) | | | (0.003) | (0.003) | (0.003) |
| TRA | | | | | | | 0.057 *** |
| | | | | | | | (0.016) |

**Table 12.** *Cont.*

|  | Model 15 | Model 16 | Model 17 | Model 18 | Model 19 | Model 20 | Model 21 |
|---|---|---|---|---|---|---|---|
| FX |  |  |  |  |  |  | 0.001 |
|  |  |  |  |  |  |  | (0.001) |
| Industry dummies | YES | YES | YES | YES | YES | YES | YES |
| _cons | 7.321 *** | 5.128 | 5.244 | 4.974 | 5.124 | 5.675 * | 2.259 |
|  | (2.278) | (3.212) | (3.230) | (3.218) | (3.264) | (3.272) | (3.453) |
| N | 1793.000 | 1027.000 | 1029.000 | 1029.000 | 1027.000 | 1027.000 | 1027.000 |
| $R^2$ | 0.261 | 0.370 | 0.367 | 0.373 | 0.363 | 0.379 | 0.401 |
| Hausman (Prob > chi2) | 0.0000 | 0.0000 | 0.0000 | 0.0000 | 0.0000 | 0.0000 | 0.0000 |

Notes: ***, and * refer to significance at 1%, 5%, and 10%, respectively; GDI has been dropped; ROE = Return on Equity; ROA = Return on Assets; TBQ = Tobin's Q; FS = Firm Size; AGE = Firm Age; TLTE = Debt Ratio; LQA = Current Asset Ratio; RDI = R&D Intensity; PER = Price-to-Earnings Ratio; PBR = Price-to-Book Value Ratio; POR = Dividend Payout Ratio; EST = Environmental Strategy; EOR = Environmental Organization; EMA = Environmental Management; EPR = Environmental Per-formance; ETO = Total Score E; SLA = Workers; SCO = Partners and Competitors; SCU = Con-sumers; SRE = Local Community; STO = Total Score S; GST = Protection of Shareholders' Rights; GBO = Board of Directors; GLI = Disclosure; GAU = Audit Body; GDI = Distribution of Management Errors; GTO = Total Score G; TRA = Trade Openness; FX = Foreign Exchange.

**Table 13.** Fixed effect panel model estimation of the impact of ESG management on Export.

|  | Model 22 | Model 23 | Model 24 | Model 25 | Model 26 | Model 27 | Model 28 |
|---|---|---|---|---|---|---|---|
| FS | 1.183 *** | 0.924 | 0.936 * | 1.065 * | 1.006 * | 1.081 * | 1.092 * |
|  | (0.222) | (0.580) | (0.560) | (0.575) | (0.600) | (0.566) | (0.570) |
| AGE | −0.033 * | 0.008 | 0.004 | −0.036 | 0.003 | 0.015 | 0.012 |
|  | (0.018) | (0.044) | (0.046) | (0.046) | (0.056) | (0.060) | (0.084) |
| TLTE | −0.001 | −0.005 | −0.004 | −0.006 * | −0.005 | −0.004 | −0.004 |
|  | (0.002) | (0.003) | (0.003) | (0.004) | (0.004) | (0.004) | (0.004) |
| LQA | 0.003 | 0.001 | −0.000 | −0.000 | −0.001 | −0.000 | −0.000 |
|  | (0.003) | (0.004) | (0.004) | (0.004) | (0.004) | (0.004) | (0.004) |
| RDI | −0.039 | −0.019 | −0.031 | −0.029 | −0.026 | −0.041 | −0.042 |
|  | (0.029) | (0.039) | (0.038) | (0.039) | (0.040) | (0.038) | (0.039) |
| PER | 0.000 | 0.000 | 0.000 | −0.000 | −0.000 | 0.000 | 0.000 |
|  | (0.000) | (0.000) | (0.000) | (0.000) | (0.000) | (0.000) | (0.000) |
| PBR | −0.028 | −0.025 | −0.023 | −0.026 | −0.025 | −0.022 | −0.022 |
|  | (0.019) | (0.031) | (0.030) | (0.031) | (0.032) | (0.030) | (0.030) |
| POR | 0.000 | 0.000 | 0.000 | 0.001 | 0.001 | 0.000 | 0.000 |
|  | (0.001) | (0.001) | (0.001) | (0.001) | (0.001) | (0.001) | (0.001) |
| EST |  |  | 0.061 *** |  |  | 0.064 *** | 0.062 *** |
|  |  |  | (0.021) |  |  | (0.022) | (0.023) |
| EOR |  |  | −0.095 *** |  |  | −0.110 *** | −0.107 *** |
|  |  |  | (0.029) |  |  | (0.030) | (0.032) |
| EMA |  |  | −0.001 |  |  | 0.003 | 0.002 |
|  |  |  | (0.007) |  |  | (0.011) | (0.012) |
| EPR |  |  | −0.000 |  |  | 0.002 | 0.002 |
|  |  |  | (0.007) |  |  | (0.008) | (0.008) |
| ETO |  | 0.013 ** | 0.009 |  |  | 0.013 | 0.013 |
|  |  | (0.006) | (0.009) |  |  | (0.010) | (0.010) |
| SLA |  |  |  | −0.000 |  | −0.001 | −0.000 |
|  |  |  |  | (0.004) |  | (0.009) | (0.009) |
| SCO |  |  |  | 0.004 |  | 0.004 | 0.004 |
|  |  |  |  | (0.004) |  | (0.005) | (0.005) |
| SCU |  |  |  | 0.004 |  | 0.002 | 0.002 |
|  |  |  |  | (0.003) |  | (0.003) | (0.003) |

**Table 13.** *Cont.*

|  | Model 22 | Model 23 | Model 24 | Model 25 | Model 26 | Model 27 | Model 28 |
|---|---|---|---|---|---|---|---|
| SRE |  |  |  | −0.010 ** <br> (0.004) |  | −0.014 *** <br> (0.005) | −0.015 *** <br> (0.005) |
| STO |  | −0.007 * <br> (0.004) |  | −0.008 * <br> (0.005) |  | −0.007 <br> (0.005) | −0.006 <br> (0.006) |
| GST |  |  |  |  | −0.000 <br> (0.007) | −0.010 <br> (0.007) | −0.010 <br> (0.008) |
| GBO |  |  |  |  | −0.001 <br> (0.012) | −0.011 <br> (0.012) | −0.010 <br> (0.012) |
| GLI |  |  |  |  | −0.001 <br> (0.005) | 0.007 <br> (0.005) | 0.007 <br> (0.006) |
| GAU |  |  |  |  | 0.000 <br> (0.006) | 0.012 <br> (0.008) | 0.012 <br> (0.008) |
| GTO |  | −0.004 <br> (0.007) |  |  | −0.003 <br> (0.008) | −0.006 <br> (0.008) | −0.006 <br> (0.008) |
| TRA |  |  |  |  |  |  | 0.016 <br> (0.041) |
| FX |  |  |  |  |  |  | −0.000 <br> (0.001) |
| Industry dummies | YES | YES | YES | YES | YES | YES | YES |
| _cons | −12.954 ** <br> (5.550) | −7.526 <br> (14.341) | −7.842 <br> (13.853) | −9.042 <br> (14.288) | −9.219 <br> (14.968) | −11.864 <br> (13.950) | −12.531 <br> (14.438) |
| N | 471.000 | 279.000 | 280.000 | 280.000 | 279.000 | 279.000 | 279.000 |
| $R^2$ | 0.121 | 0.129 | 0.199 | 0.134 | 0.089 | 0.271 | 0.272 |
| Hausman (Prob > chi2) | 0.0000 | 0.0000 | 0.0000 | 0.0000 | 0.0000 | 0.0000 | 0.0000 |

Notes: ***, **, and * refer to significance at 1%, 5%, and 10%, respectively; GDI has been dropped; ROE = Return on Equity; ROA = Return on Assets; TBQ = Tobin's Q; FS = Firm Size; AGE = Firm Age; TLTE = Debt Ratio; LQA = Current Asset Ratio; RDI = R&D Intensity; PER = Price-to-Earnings Ratio; PBR = Price-to-Book Value Ratio; POR = Dividend Payout Ratio; EST = Environmental Strategy; EOR = Environmental Organization; EMA = Environmental Management; EPR = Environmental Performance; ETO = Total Score E; SLA = Workers; SCO = Partners and Competitors; SCU = Consumers; SRE = Local Community; STO = Total Score S; GST = Protection of Shareholders' Rights; GBO = Board of Directors; GLI = Disclosure; GAU = Audit Body; GDI = Distribution of Management Errors; GTO = Total Score G; TRA = Trade Openness; FX = Foreign Exchange.

## 5. Discussion

This provides a concise and precise description of the experimental results, their interpretation, as well as the experimental conclusions that can be drawn.

### 5.1. Theoretical Implications

5.1.1. Costs and Benefits

According to the results of this study, corporate ESG management shows conflicting cost-benefit relationships according to performance types such as financial performance, market performance, and export performance. In the case of financial performance measured by ROE and ROA, all ESG management activities show a statistically significant positive (+) relationship, confirming that ESG management is an element that improves corporate performance. On the other hand, with respect to market performance, all ESG management activities were found to have no statistically significant relationship, indicating that ESG management is not a practical input for market performance. This result contradicts a series of previous studies that found a positive relationship between ESG management and market performance [115,116]. On the other hand, it can be confirmed that, unlike some ESG activities that have a positive influence on export performance, many ESG activities function as cost concepts that deteriorate export performance.

Considering these findings, it seems necessary to examine more closely whether these analysis results are due to actual strategic choices of companies or a phenomenon

in which the level or speed of regulatory and institutional development differs by ESG sector. In particular, EPR has a positive effect on financial performance (ROE), while EOR is negatively affecting export performance. This is inferred to be derived from the inherent uniqueness of export performance, which is different from general financial performance, and, thus, this study is confined to confirming the relationship between export performance and ESG management.

Furthermore, the disproportionate relationship in performance shown by the sub-components of each ESG sector requires a closer examination. In sector E, the performance implications of environmental organization (EOR), environmental strategy (EST) and environmental performance (EPR) are prominent, and in sector G, board of directors (GBO), and showed significant importance. In particular, GBO, unlike other governance elements that act as a cost concept, has been found to function as a benefit that promotes corporate and financial performance, requiring additional in-depth research on this in the near future.

### 5.1.2. Firm Size and Promotional Value versus Strategic Value

According to the results of this study, firm size has a statistically significant negative (-) relationship with financial performance such as ROE and ROA. For market performance, such as Tobin's Q, although it was not statistically significant, it showed a negative (-) relationship. Therefore, the attention must be paid to the ESG management-performance relationship moderated by the firm size, and it is necessary to consider the value of ESG management for companies, especially considering the firm size. In addition to the European Union's carbon border adjustment system, which is scheduled to be applied from 2023, the enactment of ESG supply chain management legislation which took effect in 2021 and mandates human rights and environmental due diligence, is gaining momentum as a driving force of the European Union. Furthermore, it will become a new non-tariff barrier for Korean exporters. Unlike large corporations that aim for promotional value, small and medium-sized enterprises (SMEs) consider ESG management in terms of strategic value for survival in markets.

Conglomerates with relatively ample funds and manpower can increase their value in terms of publicity by installing an ESG committee within the board of directors and publishing a sustainability report. On the other hand, the reason why SMEs need an ESG response strategy can be found in the strategic value to survive in the market economy. SMEs must come up with a plan to respond to ESG regulations that will be enforced by export destination countries. When SMEs face the implementation period of ESG principles, it will be hard for them to respond to it in a proper and timely way, so they must prepare at least 2 or 3 years in advance. A preemptive response will provide companies with strategic value for stable entry into the export market. In particular, the decrease in exports of domestic companies in the global market, where their market share is declining due to intensifying competition, makes us more concerned about the non-tariff barrier effect that ESG-related regulatory policies will bring.

> "*The reason why we prepare ourselves even though our trading partners do not ask for it derives from the fact that we believe that the time when ESG regulations will realistically come soon . . . If we start responding when ESG regulations start then it will be too late . . . We need to prepare from now on for 2 or 3 years in advance. This is because it can be implemented right away when a realistic problem arises three years later . . . Companies that are not prepared will only start preparing at that time, and then our company will have the ability to respond immediately, so it will have relatively [high] market competitiveness.*" [117]

### 5.2. Managerial Implications

In the case of Korean companies, after integrating the ESG evaluation information of 1292 companies, the ESG overall rating shows a continuous downward trend that starts to rise again from 2018. In the year 2021, the average of the comprehensive grade is a B, ranging between B and B+. The environmental category ratings of all companies continued

to decline until the year 2020, then rebounded in the year 2021, recording higher scores than in the year 2019. Social domain ratings showed a downward trend until the year 2015, but then steadily increased until the year 2020. In the case of the corporate governance category rating, it recorded a higher score than the other areas, and since the year 2015 the average rating has shown an upward trend. In the year 2021, it can be seen that it is close to the average B+ rating. Looking at the distribution of ESG ratings, it can be seen that the largest number of companies are concentrated in the C and B grades. This seems to be the reason why the average grade does not rise further from the B grade.

Contrary to the low level of environmental scores, management activities in the environmental sector were found have a positive effect on financial performance, market performance, and export performance of companies. However, regarding export performance, it can be seen that management activities in the environment and governance sector have a negative impact and act as a cost concept. Meanwhile, unlike the social sector score showing the highest level, management activities in the social sector did not have a statistically significant effect on the company's financial performance, market performance, and export performance.

Given that export expansion generally has a positive effect on a company's market evaluation, the negative impact of ESG management on exports found in this study provides an interesting practical implication. An export market cannot be established in a short time due to institutional barriers, technical barriers, and physical barriers. In addition, a firm's entry into the export market is determined by various factors such as monopolistic firm-specific advantages [118,119] and liabilities of foreignness [120–122]. Although regionally limited to the case of Korean companies, the results of this study imply that ESG management has not yet functioned as a competitive advantage for export competitiveness. However, this result does not suggest absolute strategic implications for companies' internationalization strategies.

From these practical characteristics of the ESG management of Korean companies, the impact of ESG management on financial performance, market performance, and export performance is disproportionate. From these findings, we came to the conclusion that it is necessary to promote corporate environmental, social, and corporate governance management in a balanced manner by considering the diversity of corporate performance indicators. On the other hand, the meaninglessness of ESG management shown in market performance indicator models such as Tobin's Q suggests that corporate ESG management at the domestic level has not yet manifested a signal effect in terms of socially responsible investment (SRI) [115,116]. Nevertheless, it is worth paying attention to the fact that major global financial institutions such as BlackRock (2020) have recently declared that they will decide the scope of investment according to ESG management for Korean companies [123].

From the resource-based point of view (RBV), which is often mentioned in the strategic management dimension, a company's ESG activities are valuable, scarce, inimitable by competitors, and sustainable competitive advantage resources that are easy to embody in organizations in seizing market opportunities and eliminating threats [124]. In addition, corporate ESG management realizes "relational immanence" by building interaction and trust among members of the organization, and "social capital" by forming "structural immanence" that is fixed through knowledge sharing and productive communication [125].

What matters is that the values and sensitivities that companies pursue for ESG activities are different by industry. The food industry is relatively sensitive to activities in the environmental sector, while the medical device industry is more sensitive to social and governance than environmental (see interview below). Therefore, it is important for individual companies to strategically promote ESG management centered on sectors with high sensitivity according to the attributes of their industries. What individual companies should pay attention to here is the fact that they must choose whether to pursue ESG activities with a focus on balance or sequentially, considering the strategic fit of the company.

*"In the case of the medical device sector, which does not use a lot of industrial water, the E sector has relatively little meaning in ESG evaluation . . . There is no big difference between companies because they follow the standard, and since they use a lot of industrial water, they are very sensitive to the environment . . . ."* [117]

## 6. Conclusions

Until now, we have examined the relationship between corporate ESG management activities and corporate performance and found the following key facts. First, as expected from this study, unlike the positive or negative statistical significance of the relationship between general financial attributes and corporate financial performance (ROE, ROA), ESG management shows a statistically significant relationship with corporate financial performance. All hypotheses were not accepted, except for environmental factors showing high sensitivity with the environmental sector. Second, ESG management scores, which are used as financial market investment indicators, did not show a statistically significant relationship with market performance; i.e., Tobin's Q. It contrasts to the partial adoption of the hypotheses shown in its relation to corporate financial performance. Third, the ESG management showed a clear relationship with export performance in sectors E and S, but there was no statistically significant relationship with sector G and a relatively high relationship with sector E, resulting in an imbalance between ESG sectors. From this series of main findings, we can derive theoretical implications that the performance implications of ESG management vary according to the type of corporate performance. These findings are expected to provide practical implications for performance management in the decision-making of corporate stakeholders regarding ESG investment activities.

Unlike these key findings, this study does not tell us the full story of the relationship between ESG management and corporate performance. This is because there are still future research themes that have not been addressed due to the methodological limitations of this study and potential errors in the data used. First of all, considering the disproportionate aspects of the ESG management and performance relationship, further research is needed to examine whether the balance between and among ESG sectors or their sequential progress is important for better performance. As in strategic choice studies on exploitation and exploratory activities [126–128], a firm's strategic choice always follows the next order after identifying what options are available. When these possible items coexist, we face the issue of which one to choose first or at the same time. The author expects that the balanced implementation of environmental, social, and corporate governance management will have a positive (+) impact on the company's financial performance, market performance, and export performance.

Second, depending on the characteristics of each industry, the extent to which companies invest scarce resources and capabilities in management activities by ESG sector may vary. This is because the sensitivity of each sector is different or the level of regulation is different for each sector. In the same context, Na and Hong (2011) analyzed differences in corporate value according to types of detailed activities that make up CSR activities, such as soundness and fairness of corporate activities, social service activities, consumer protection activities, employee satisfaction, environmental protection activities, contribution to economic development, etc. Finally, it is also interesting to compare the relationship between corporate ESG management and corporate performance by dividing firms into non-financial and financial groups [53]. Some previous studies show limitations in not considering the fact that the performance implications of corporate ESG management differ depending on the diversity of companies. As with the hypotheses discussed above, it is also possible that the industry has an effect on different relationships and various values [53].

The relationship between corporate ESG management and performance can be complete only when future research themes are covered. Hopefully, this study will serve as a starting point for stimulating valuable future research by presenting basic perspectives and discussion topics for analysis.

**Author Contributions:** Conceptualization, O.-S.Y. and J.-H.H.; methodology, O.-S.Y.; writing and editing, J.-H.H.; funding acquisition, J.-H.H. All authors have read and agreed to the published version of the manuscript.

**Funding:** This research was supported by Hallym University Research Fund, 2019 (HRF-201911-006).

**Institutional Review Board Statement:** Not applicable.

**Informed Consent Statement:** Not applicable.

**Data Availability Statement:** All data generated or analyzed during this study are available from the corresponding authors upon reasonable request.

**Acknowledgments:** The authors would like to thank KCGS for the data collection and thank the anonymous reviewers and editor.

**Conflicts of Interest:** The authors declare no conflict of interest.

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
