# Peer review of "Assessing the Effect of Corporate ESG Management on Corporate Financial & Market Performance and Export"

_sustainability, doi:10.3390/su15032316_

Round 1
Reviewer 1 Report
This paper is meaningful in that it attempted to confirm the various aspects of Korean esg management on performance. In addition, this paper is differentiated in that it attempted to increase the rigor of the analysis by utilizing various variables and methodologies. The hypotheses presented by the authors and the results of their empirical analysis are also quite interesting, and the implications of these results are also meaningful.
Author Response
Many thanks for your marvelous comments. Despite your positive comments, we revised some parts of the manuscript accepting critical comments from other reviewers as well as the authors' findings for systemic errors. Please find revised manuscript for confirming the better quality of the manuscript.
Reviewer 2 Report
The topic of this paper is interesting, and with potential to dive deep inside. Here are some comments and recommendations to be addressed as follows:
1. The abstract should include the selected sample size.
2. Include Korea as a key word.
Introduction
3. Long paragraphs are not a good standard to enhance the readability of the paper.
4. More emphasis on the most important research papers in the research area that need to be relevant for explaining the keywords from the objective would be advantageous to the reader.
5. The research gap and research motivation are not clear in the paper.
6. In the last part of the introduction, develop a few lines briefly presenting the methodology used in your study, the main results, and the theoretical and managerial contributions of your study.
Theoretical background
7. Theoretical background presents some interesting ideas; however, it would be better if supported with a more recent literature review. More emphasis on the most important research papers in the research would be advantageous to the reader.
8. In (Line 98), the authors mentioned “Triple profit”. However, the correct concept developed by Elkington (1990) is “the triple bottom line”.
9. Lack of references and sources of statistical information in some parts of the article, for example in (Line 110) the authors stated that “There are more than 120 ESG evaluation institutions around the world”. What is the source of this information?
10. The choice of the 8 financial attributes must be explained based on the previous literature.
11. Each financial attribute should be more developed. The authors stated only the definition of each attribute while the relationship (positive or negative or not clear in the literature) between each attribute and the firm performance must be clearly explained and then they mention the hypothesis.
12. The bloc of hypotheses (hypotheses through 1 to 2-3) should be avoided. Mention each hypothesis after its development.
13. Make hypotheses through 2-1 to 2-3 shorter. Avoid unnecessary explanation in the hypothesis.
Methodology
14. The sub-section 3.1 should be divided into two sub-sections: Add a first sub-section (3.1.) to explain sample selection (initial sample, excluded firms, final sample size and period of the study) and data collection source for ESG data, performance measure and control variables measures. Sub-section (3.2) will be to only explain the research model.
15. The authors have to explain the methodology used. Why they tested 3 regressions?
16. The authors have to mention the source of independent, predictor, and control variables’ measurements. Mention all studies used as base to select these measurements
17. What is the correlations matrix used for the multicollinearity test? Pearson or Spearman ?
18. Re-present the table of correlation matrix (landscape orientation) to be easily and clearly read.
19. Explain more the empirical findings (estimation results). Develop the findings and compare them with previous results found in the literature.
Conclusions
20. Conclusions can be enhanced if the research limitations and future research directions are highlighted.
Other important comments:
21. The following paper are recommended to include in the next version:
- Kouaib, A. (2022). “Corporate Sustainability Disclosure and Investment Efficiency: The Saudi Arabian Context”
- Kouaib, A., Bouzouitina, A., and Jarboui, A. (2021). “CEO behavior and sustainability performance: the moderating role of corporate governance”.
- Kouaib, A., Mhiri, S., and Jarboui, A. (2020). “Board of directors’ effectiveness and sustainable performance: The triple bottom line”
22. Line (74): “The authors” instead of “the author”.
23. A consistent English language proofing needs to be performed.
Author Response
Many thanks for your critical and gentle comments for our manuscript. We think that this manuscript has developed with your critical comments. We found that there were missing information and some systemic errors in the analysis presentation. Even we found that your comments are very significant for confirming the results. We attached the response letter in which you can find what we revised and how to rearranged the contents of the paper. The modified and revised parts are marked in blue text. (please See attached file)

Reviewer 3 Report
ID_2150520: Title: Assessing the Effect of Corporate ESG Management on Corporate Financial & Market Performance and Export
Dear authors and editor,
This article focuses on the comparative relationship of ESG performance and financial performance (both market and book value based) and exports for Korean dataset.
I can some aspects of the paper that should be improved for its consideration.
First of all the title of the paper is misleading. (Tobin’s Q is not a market performance measure). Moreover, from the ESG score provided by the independent agencies cannot be used as a proxy for ESG management.
Abstract is not clear. Especially the way of reporting findings. Further, authors should make sure that they have used a time series data or panel data. From methodology section it seems the panel data.
In the introduction section, Contribution of the study is not clear. Please clarify your research questions, objectives, background motivation, theoretical and empirical motivation and the lines of contributions to the literature. You can do this by sharply articulating your research questions/objectives, identify the potential theoretical, background and theoretical motivation or gaps, and explain how your study contributes to the literature. You can do this by highlighting the weaknesses of prior studies as well. Currently, your introduction is very dry. Additionally, you need state clearly the contributions of the paper. For example, "Consequently, the current paper seeks to make the following contributions to the existing literature. First,…, Second,…., Third, …, Fourth,… and so on". The description of the contribution needs to be more forensic, needs to be more focussed.
In the literature review and hypothesis development section, I suggest authors to delete the section 2.2, as the relationship between the different firms level characteristics such as firm size and corporate performance has already been established. It does not add anything new.
There is very significant literature on ESG, the authors need to update the literature and include recent published papers (2021-2022) and they need to discuss the research gaps and then they need to explain how the current paper fills at least one of these gaps.
The authors show also base there hypothesis on theoretical foundation: What is the underlying theory that leads to the development of the hypotheses? The authors need to enhance their hypotheses development by: (i) drawing on the theory; (ii) empirical literature; (iii) research setting/contextual insights; and (iv) then setting up their hypotheses. They will do this for each hypothesis. Currently, They have not developed your hypotheses in this way. They will need to so by drawing on both seminal (old) and recently (newly) published studies.
The main contribution of the study is the export or internationalization factor, authors should focus more on this part.
There needs to be a more comprehensive and formal discussion methodology used to conduct this study. Why there is need of reduced models, or full model need to be explained. Moreover, the symbols used are confusing, E and Eit total.. How these are different. Need to describe model in more details.
Tables presented are not readable, for instance, there is no need for table 1, i recommend deleting it or move to the appendix. In table 2 definitions authors should add a column about the source (from where they collected the data for each variable). Table 4 is not readable… I recommend authors to focus on the key variables correlation and report only those(they can skip the control variables from correlations or report them in the appendix).
There is also no need to report the results of unit root tests. So no need of table 5. Authors should only describe in text that the pre regression tests are ok. Moreover, it is not clear from the table 5 that which unit root results it is reporting.
Table 6-9 again, i recommend the authors delete the section about the relationship of firm level financial variables with dependent variable, use it only as control variables. Moreover, R-square values for the table 6 and 7 very low, showing there is some issue for model fitness. So, authors should explain it.
The authors need to link their findings more strongly to the: (i) theory, (ii) empirics, (iii) context of region; and (iv) highlight their economic, academic/research and policy implications. In the discussion of the results please focus on the novel findings and insights vis-à-vis the existing literature.
In the conclusion, the authors need to expand the discussions relating to implications, limitations and avenues for future research.
Author Response

(The authors gave the same response as above.)

Round 2
Reviewer 2 Report
The revised version is now suitable for publication. Good luck for the authors.
Author Response
We would like to thank reviewers for very kind comments and the second response to the revision of the manuscript by accepting the revised manuscript. Additionally, we revised 1) some technical errors concerning duplicate references, (Line 907-1141) as a result the no. of references reduced to 128 (previous 165 no.) 2) replaced the first ‘Korea’ with ‘South Korea (hereafter Korea)’, (Line 35) and 3) removed the dropped variable in the result table accompanied by describing this thing in Table Note “GDI has been dropped”. (Line 679, 689, 699, 709) The modified and revised parts are marked in green text, accompanied by the previous revised text, which are marked in blue and red text.

Reviewer 3 Report
Thank you for incorporating suggested changes.
Author Response

(The authors gave the same response as above.)
